# Topograph: An efficient Graph-Based Framework for Strictly Topology Preserving Image Segmentation

**Laurin Lux**[*1,3,5], **Alexander H. Berger**[*1,5], **Alexander Weers**[1], **Nico Stucki**[1,3,4],
**Daniel Rueckert**[1,2,3,4], **Ulrich Bauer**[1,3,4], **Johannes C. Paetzold**[2,5]
[1]School of Computation, Information and Technology, Technical University of Munich, DE
[2]Department of Computing, Imperial College London, UK
[3]Munich Center for Machine Learning, DE
[4]Munich Data Science Institute, Technical University of Munich, Munich, DE
[5]Weill Cornell Medicine, Cornell University, New York City, NY, USA
{laurin.lux,a.berger}@tum.de; jpaetzold@med.cornell.edu

## Abstract

Topological correctness plays a critical role in many image segmentation tasks, yet most networks are trained using pixel-wise loss functions, such as Dice, neglecting topological accuracy. Existing topology-aware methods often lack robust topological guarantees, are limited to specific use cases, or impose high computational costs. In this work, we propose a novel, graph-based framework for topologically accurate image segmentation that is both computationally efficient and generally applicable. Our method constructs a component graph that fully encodes the topological information of both the prediction and ground truth, allowing us to efficiently identify topologically critical regions and aggregate a loss based on local neighborhood information. Furthermore, we introduce a strict topological metric capturing the homotopy equivalence between the union and intersection of prediction-label pairs. We formally prove the topological guarantees of our approach and empirically validate its effectiveness on binary and multi-class datasets. Our loss demonstrates state-of-the-art performance with up to fivefold faster loss computation compared to persistent homology methods.[1]

## 1 Introduction

In segmentation and structural analysis tasks, maintaining topological integrity is often more critical than simply improving pixel-wise accuracy. For example, in medical imaging, the topological integrity of segmented structures, such as blood vessels Todorov et al. (2020) or neural pathways, can be crucial for accurate diagnosis and functional analysis (Briggman et al., 2009). However, topological errors, such as loss of connectivity, are common in practice, even when pixel-wise accuracy is high. Standard pixel-based loss functions, such as Dice-loss, do not adequately address these issues. While they minimize pixel-level discrepancies, they do not take into account changes in topology, which may be caused by few or even single pixels. As a result, even small pixel-wise errors can lead to significant topological failures.

Previous works have shown how different topology-aware methods can improve the integrity of target structures without sacrificing pixel-wise accuracy. Task-specific methods, such as those designed for tubular structure segmentation (Shit et al., 2021; Kirchhoff et al., 2024), are computationally efficient and perform well in their respective domains. However, they do not generalize effectively to other types of topological structures or datasets. In contrast, persistent homology (PH)-based methods can provide strong theoretical guarantees and deliver state-of-the-art performance (Hu et al., 2019; Stucki et al., 2023; Clough et al., 2020), but are computationally more demanding. Other topology-aware methods can be more versatile and computationally efficient, but lack theoretical guarantees for topological correctness (Mosinska et al., 2018; Funke et al., 2018; Hu et al., 2021).

---

[*]Equal Contribution
[1]Code is available at https://github.com/AlexanderHBerger/Topograph

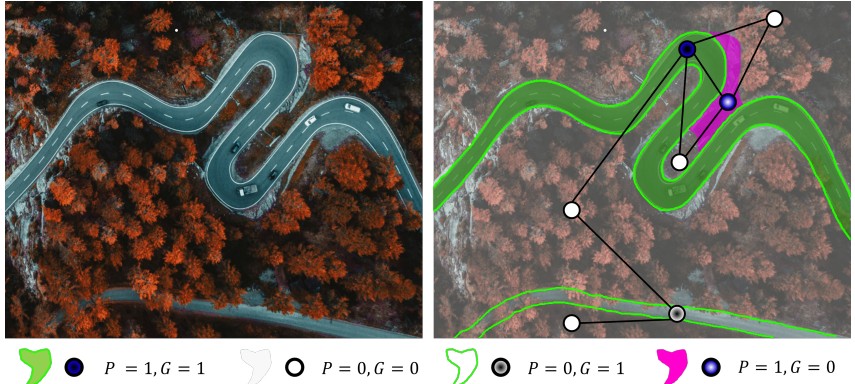

Figure 1: Visualization of the proposed component graph representation. Left: Input image; Right: Overlay of the prediction ($P$) and ground truth ($G$). The bright green lines indicate the foreground structures in the ground truth, with (darker) green regions indicating correctly predicted foreground and pink regions representing incorrectly predicted foreground. A combined component graph $\mathcal{G}(P, G)$ is constructed to efficiently identify topological errors, which are used to compute a loss.

This work proposes a loss function that generalizes to various segmentation tasks where topology is crucial. Our method is based on a component graph that combines joint topological information of ground truth and prediction (see Figure 1). A theoretically founded analysis of the nodes in the graph allows the identification of topologically critical regions, which we then use for loss computation.

**Our contribution.**  We (1) establish a metric that captures topological correctness with strict theoretical guarantees, especially capturing the homotopy equivalence between union and intersection of a label/prediction pair, and (2) formulate a general topology-preserving loss for training arbitrary segmentation networks. Specifically, our loss formulation

1. surpasses existing methods in terms of topological correctness of predictions;
2. provides stricter topological guarantees than existing works, i.e., formal guarantees beyond the homotopy equivalence of ground truth and segmentation, by extending the enforced homotopy equivalence to their union and intersection through the respective inclusion maps, capturing the spatial correspondence of their topological properties;
3. is time and resource-efficient because of its low asymptotic complexity ($O(n \cdot \alpha(n))$) and empirically low runtime;
4. and is flexible, making it applicable to arbitrary structures and image domains.

We empirically validate the prediction performance on various public datasets for binary and multi-class segmentation tasks.

**Related Work**  Significant progress has been made in segmentation methods that preserve topological accuracy. The use of PH-based loss functions for training segmentation networks (Hu et al., 2019; Clough et al., 2020) ensures global topological correctness by aligning Betti numbers when minimized to zero. However, two issues persist: 1) most methods cannot guarantee spatially related matched structures, and 2) computational cost. Stucki et al. (2023) introduce the Betti Matching concept, which ensures spatially correct barcode matching. However, the cost of barcode computation is substantially higher compared to overlap-based methods, making its derived methods applicable only to relatively small patch sizes up to $80 \times 80$. This limits the applicability to medical and natural images, where whole images are commonly an order of magnitude larger. Another limitation is the gradient's dependence on just two simplex values, which Nigmetov & Morozov (2024) recently showed to hinder optimization speed. While other methods are computationally more efficient (Hu et al., 2021; Mosinska et al., 2018), they offer limited guarantees of topological correctness. Task-specific, overlap-based approaches have been proposed for tubular structures where connectivity is the key topological feature. ClDice (Shit et al., 2021) calculates a loss term based on the union of ground truth skeletons and predicted volumes, a method extended in recent studies (Kirchhoff et al., 2024; Menten et al., 2023). Other approaches refine tubular-structure features through iterative feedback learning strategies (Cheng et al., 2021) or rely on post-processing networks (Li et al., 2023;

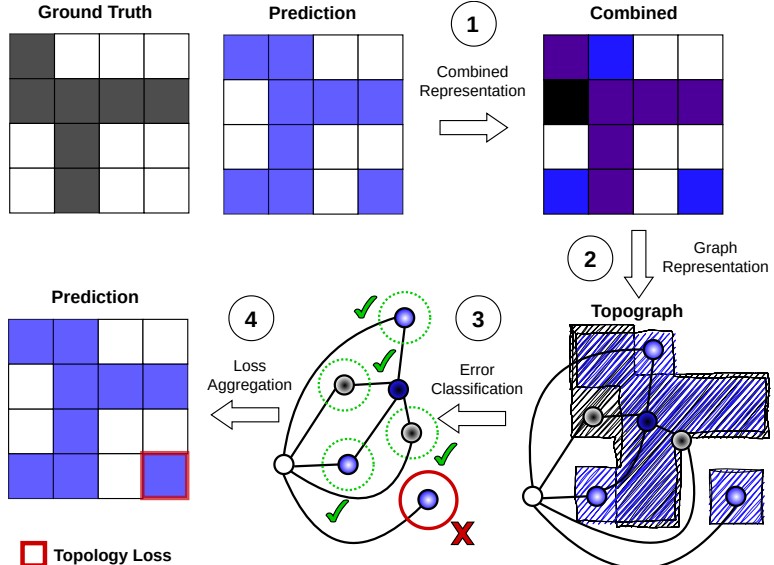

Figure 2: Overview of our proposed method. (1) We use the prediction in each iteration of the training phase to build a combined image with the labels. (2) Based on the combined image, we construct a superpixel graph $\mathcal{G}(P, G)$ that encodes the full topological information of both segmentations. (3) We can identify topologically relevant errors using each node's local neighborhood. (4) Finally, we can backtrack the critical errors to image regions and calculate a topological loss function. This allows an efficient formulation of a topological loss with strict theoretical guarantees.

Wu et al., 2024). In neuroscience, prior work presented various segmentation methods minimizing *false split* and *false merge* errors in neuron segmentation tasks (Funke et al., 2018; Briggman et al., 2009). These errors are related to topological segmentation errors in dimension 0. Accordingly, Funke et al. (2017) presented the TED metric measuring the number of these errors outside a pre-defined "tolerance" region. However, none of these approaches generalize effectively to arbitrary structures and complete topological information.

## 2  METHOD

We propose a topology-preserving loss function based on the *combined component graph* $\mathcal{G}(P, G)$, which encodes topological information from both the label and prediction. Using this graph, we implement an algorithm for identifying regions in the prediction that must be corrected to adhere to the ground truth topology. Thereby, we aim to optimize the network such that topologically critical regions are correctly predicted while less significant regions are ignored. To construct the component graph, we pair the prediction $P$ and ground truth $G$ into a 4-class image $C$ that is then partitioned into superpixels via connected component labeling. Each superpixel, composed only of pixels with the same class in $C$, corresponds to exactly one node in the resulting component graph $\mathcal{G}(P, G)$. This planar and bipartite graph, with edges connecting adjacent superpixels, captures the topology of both $P$ and $G$. Analyzing the local neighborhood of each node in $\mathcal{G}(P, G)$, we can identify *critical nodes* that represent incorrectly predicted regions causing topological errors. Formally, the set of critical nodes is defined by all incorrectly predicted nodes that do not have exactly one correctly predicted foreground and one correctly predicted background neighbor. The final loss function is obtained by pixel-wise aggregation for each critical region. A schematic overview of the method from graph generation to loss calculation is provided in Figure 2. In the following sections, we provide a detailed description of our method, its theoretical guarantees, its asymptotic complexity, and some interesting adaptations to the method.

### 2.1  COMPONENT GRAPHS ENCODING DIGITAL IMAGE TOPOLOGY

Our method relies on the fact that a *component graph* $\mathcal{G}(I)$ of a binarized segmentation encodes the relevant topological information of an underlying 2D segmentation. The vertices $\mathcal{V}(\mathcal{G}(I))$ resemble the connected components of foreground and background, whereas the edges $\mathcal{E}(\mathcal{G}(I))$ encode the neighborhood information of these components.

We model the topology of a binarized digital image $I \in \{0, 1\}^{h \times w}$ by a two-dimensional cubical grid complex $C = [0, h] \times [0, w] \subset \mathbb{R}^2$ using the T-construction, i.e., a voxel $(i, j) \in \{1, \ldots, h\} \times \{1, \ldots, w\}$ corresponds to a top-dimensional cell $[i - 1, i] \times [j - 1, j] \in C$. To apply duality arguments and exclude edge cases, we add an additional background cell $c_\star$ that is attached to the boundary $[0, m] \times \{0, n\} \cup \{0, m\} \times [0, n]$ of $C$. This turns the cubical grid complex $C$ into a CW-complex $\widetilde{C}$ that is homeomorphic to the sphere $S^2$.

The *foreground* $F(I)$ is given by the closure of the union of 2-cells whose voxels take value 1 and its *background* $B(I)$ is given by the complement $\widetilde{C} \setminus F(I)$. Foreground and background decompose into connected components $F_1, \ldots, F_k$ and $B_1, \ldots, B_l$, which together form the vertices of its *component graph* $\mathcal{G}(I)$. The component graph is a bipartite tree with edges between a foreground component $F_i$ and a background component $B_j$ if and only if $F_i \cap \overline{B_j} \neq \emptyset$. Note that the component graph determines the homotopy type of the foreground $F(I)$, since its Betti numbers can be inferred from it. We find $b_0(F(I))$ to be the number of foreground vertices of $\mathcal{G}(I)$ and

$$b_1(F(I)) = \sum_{i=1,\ldots,k} \deg_{\mathcal{G}(I)}(F_i) - 1,$$

where $\deg_{\mathcal{G}(I)}(F_i)$ denotes the number of edges in $\mathcal{G}(I)$ that contain vertex $F_i$. Beyond that, starting from the surrounding background node along a path to a leaf of $\mathcal{G}(I)$ informs about the relationship of consecutive nodes of the path. A background component following a foreground component is a hole within the previous foreground component.

We consider the *thickened foreground* $F_\epsilon(I) = D_\epsilon(F(I))$ and the *thinned out background* $B_\epsilon(I) = \widetilde{C} \setminus F_\epsilon(I)$ to prevent connectivity issues in the *combined component graph*. Here, for a subset $X \subseteq \mathbb{R}^2$, we denote by $D_\epsilon(X) = \{y \in \mathbb{R}^2 \mid \exists x \in X : \|x - y\|_\infty \leq \epsilon\}$. Note that $F_\epsilon(I)$ deformation retracts onto $F(I)$ and $B(I)$ deformation retracts onto $B_\epsilon(I)$.

**Combined component graph**    Based on an overlay of a binarized prediction $P \in \{0, 1\}^{h \times w}$ and its ground truth segmentation $G \in \{0, 1\}^{h \times w}$, we can cover the CW-complex $\widetilde{C}$ by four subspaces:

1. $\mathrm{TP} = F_\epsilon(P) \cap F_{2\epsilon}(G)$ (*true positive*),
2. $\mathrm{TN} = B_\epsilon(P) \cap B_{2\epsilon}(G)$ (*true negative*),
3. $\mathrm{FN} = B_\epsilon(P) \cap F_{2\epsilon}(G)$ (*false negative*),
4. $\mathrm{FP} = F_\epsilon(P) \cap B_{2\epsilon}(G)$ (*false positive*).

Each of these subspaces decomposes into connected components $\mathrm{TP}_1, \ldots, \mathrm{TP}_k$, $\mathrm{TN}_1, \ldots, \mathrm{TN}_l$, $\mathrm{FN}_1, \ldots, \mathrm{FN}_m$, $\mathrm{FP}_1, \ldots, \mathrm{FP}_n$, which form the vertices of the *combined component graph* $\mathcal{G}(P, G)$. Furthermore, we add an edge between two vertices of $\mathcal{G}(P, G)$ if and only if their closures intersect in an infinite set of points, i.e., the intersection contains a 1-dimensional cell.

The combined component graph $\mathcal{G}(P, G)$ combines the information of ground truth and predicted segmentation. Since we use the thickened foregrounds and thinned out backgrounds (for visualization, see Figure 7), $\mathcal{G}(P, G)$ is a bipartite graph whose edges only occur between vertices contained in $\mathrm{T} = \mathrm{TP} \cup \mathrm{TN}$ and $\mathrm{F} = \mathrm{FP} \cup \mathrm{FN}$. Figure 2 visualizes $\mathcal{G}(P, G)$ (bottom right) for a given prediction and ground truth segmentation (top left). The component graphs $\mathcal{G}(P)$ and $\mathcal{G}(G)$ are quotients of $\mathcal{G}(P, G)$ that can be obtained by contracting all edges incident to nodes with the same label with respect to the respective image. Within $\mathcal{G}(P, G)$, it is possible to identify critical components of the prediction that represent topological errors in the segmentation.

## 2.2 IDENTIFYING TOPOLOGICALLY CRITICAL COMPONENTS

Given $\mathcal{G}(P, G)$, our goal is to find a set $\mathcal{V}_c$ of *critical* vertices, whose pixels must be adjusted to guarantee the same homotopy type for prediction and ground truth. Correcting the complete set of incorrectly classified vertices $\mathcal{V}_F = \{v \in \mathcal{V}(\mathcal{G}(P, G)) : v \subseteq \mathrm{F}\}$ forces equality of ground truth and prediction. However, a loss that is defined based on this relabeling is similar to a pixel-wise loss and does not focus on topologically critical components over topologically irrelevant components. Therefore, we aim to identify all vertices $\mathcal{V}_r \subseteq \mathcal{V}_F$ whose labeling is irrelevant for the homotopy type of the prediction and exclude them from $\mathcal{V}_c$ to emphasize the importance of topological errors. In $\mathcal{G}(P, G)$, those vertices can be characterized as vertices that have exactly one correctly classified foreground neighbor and exactly one correctly classified background neighbor. We define the

*regular* vertices

$$\mathcal{V}_r = \{v \in \mathcal{V}_{\mathrm{F}} \mid (\exists! \, s \in \mathcal{N}_{\mathcal{G}(P,G)}(v) : \, s \subseteq \mathrm{TP}) \wedge (\exists! \, t \in \mathcal{N}_{\mathcal{G}(P,G)}(v) : \, t \subseteq \mathrm{TN})\}. \quad (1)$$

Here, $\mathcal{N}_{\mathcal{G}(P,G)}(v)$ denotes the neighboring vertices of $v$ in $\mathcal{G}(P, G)$. We provide an intuition for this definition in the Supplementary Material (see Figure 8).

### 2.3 COMPONENT GRAPH LOSS

After excluding the regular vertices $\mathcal{V}_r$, which are incorrectly classified but irrelevant for the topology of the prediction, the set of critical vertices is given by

$$\mathcal{V}_c = \mathcal{V}_{\mathrm{F}} \setminus \mathcal{V}_r \quad (2)$$

and a loss can be created from the remaining incorrectly classified vertices in this set.

We calculate the loss of a single vertex as the average score of the predicted class among all its pixels ($\bar{y}_v$). The combined loss is the sum of all loss terms from the individual regions. Notably, the loss aggregation for each vertex in $\mathcal{V}_r$ remains a design choice and can be adjusted to the particularities of any target task. With the design choices described above, the loss $\mathcal{L}_{CG}$ can be denoted as

$$\mathcal{L}_{CG} = \alpha \sum_{v \in \mathcal{V}_c} \bar{y}_v \quad (3)$$

where $\alpha$ is a factor to balance $\mathcal{L}_{CG}$ with a pixel-wise loss term. In this formulation, all pixels that constitute the component of a vertex contribute to the loss via their class score.

### 2.4 TOPOLOGICAL GUARANTEES OF THE COMPONENT GRAPH LOSS

By definition, our loss formulation drops to zero if and only if the set of critical vertices is empty, either because no misclassified vertices remain or because all misclassified vertices in $\mathcal{G}(P, T)$ are regular. We obtain the following proposition

**Proposition 2.1.** *If $\mathcal{L}_{CG}(P, G) = 0$, the commutative diagram consists of deformation retractions:*

$$
\begin{array}{ccc}
 & F_\epsilon(P) & \\
\overset{\simeq}{\nearrow} & & \overset{\simeq}{\searrow} \\
F_\epsilon(P) \cap F_{2\epsilon}(G) & & F_\epsilon(P) \cup F_{2\epsilon}(G) \\
\overset{\simeq}{\searrow} & & \overset{\simeq}{\nearrow} \\
 & F_{2\epsilon}(G) &
\end{array}
$$

We sketch a proof of the fact that the inclusion $F_\epsilon(P) \cap F_{2\epsilon}(G) \hookrightarrow F_\epsilon(P)$ is a deformation retraction. The following proof by induction can be applied similarly to the other inclusions.

*Sketch of proof.* First note that $F_\epsilon(P) \cap F_{2\epsilon}(G) = \mathrm{TP}$ and $F_\epsilon(P) = \mathrm{TP} \sqcup \mathrm{FP}$. We will prove the statement by induction on the number of regular vertices in $\mathcal{G}(P, G)$ that are contained in FP.

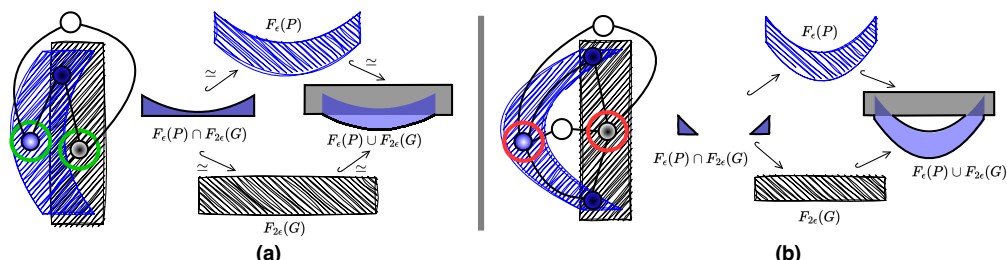

Figure 3: Inclusion diagrams for two exemplary prediction label pairs (a) and (b). For example in (a), all the inclusions are homotopy equivalences, which corresponds to the absence of critical nodes in $\mathcal{G}(P, G)$. In example (b), none of the inclusions are homotopy equivalences, which corresponds to the presence of critical nodes in $\mathcal{G}(P, G)$.

For 0 regular vertices, $F_\epsilon(P) \cap F_{2\epsilon}(G) = F_\epsilon(P)$ holds and $F_\epsilon(P) \cap F_{2\epsilon}(G) \hookrightarrow F_\epsilon(P)$ is trivially a deformation retraction. So assume that FP contains $n > 0$ regular vertices, $\text{FP}_1, \ldots, \text{FP}_n$. By induction hypotheses, the space $X = \text{TP} \sqcup (\bigsqcup_{i=1,\ldots,n-1} \text{FP}_i)$ deformation retracts onto TP. Hence, it remains to show that $\text{TP} \sqcup \text{FP}$ deformation retracts onto $X$. Since we consider thickened foregrounds and thinned out backgrounds, the closure of any connected component of FP is homeomorphic to a closed disk $D^2$ with finitely many open balls $B^2$ cut out. By our assumption that $\text{FP}_n$ is regular, it has exactly one neighbor $\text{TP}_i$ contained in TP and exactly one neighbor $\text{TN}_j$ contained in TN. Furthermore, it cannot have any additional neighbors, since $\mathcal{G}(P, G)$ is bipartite. Therefore, it has exactly two neighbors, which are connected subsets of $\mathbb{R}^2$, and we conclude that it can be at most one ball that is cut out of the disk. We distinguish two cases:

*Case 1*: In case the closure of $\text{FP}_n$ is homeomorphic to $D^2$, its boundary is homeomorphic to $S^1$ and is divided into two connected parts: a shared boundary with $\text{TP}_i$ and a shared boundary with $\text{TN}_j$. This follows by connectedness of $TP_i$ and $TN_j$. Hence, without loss of generality, we can assume that the northern hemisphere of $S^1$ is the shared boundary with $\text{TP}_i$ and the southern hemisphere is the shared boundary with $\text{TN}_j$. Then clearly $\text{TP} \sqcup \text{FP}$ deformation retracts onto $X$ by pushing the shared boundary of $\text{FP}_n$ with $\text{TN}_j$ along the disk onto the shared boundary of $\text{FP}_n$ with $\text{TP}_i$.

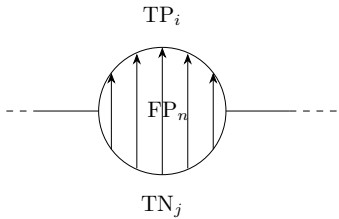

*Case 2*: In case the closure of $\text{FP}_n$ is homeomorphic to an annulus $D^2 \setminus B^2$, its boundary is homeomorphic to two copies of $S^1$. One of those is the shared boundary with $\text{TP}_i$, and the other one is the shared boundary with $\text{TN}_j$. This time, we can push the shared boundary with $\text{TN}_j$ through the annulus to the shared boundary with $TP_i$ to see that $\text{TP} \sqcup \text{FP}$ deformation retracts onto $X$. $\qquad\square$

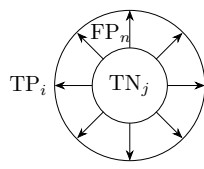

As an immediate consequence of this deformation retractions, $\mathcal{L}_{CG} = 0$ implies that the *BM error* (Stucki et al. (2023)) is $0$ too. This is because homotopy equivalences induce isomorphisms in homology and, therefore, the obtained *induced matchings* match identical intervals of the respective barcodes. Therefore, $\mathcal{L}_{CG}$ is a more sensitive loss function than the *BM loss*. Similar to *Betti Matching (BM)*, our approach does not only consider the topological spaces represented by images but also takes the natural inclusions that connect them into account. While Betti Matching uses induced matchings of persistence barcodes to compare prediction and label with respect to the inclusions into their union, $\mathcal{L}_{CG}$ also considers the inclusions of their intersection, see Figure 3 and Supplementary Figure 11.

## 2.5 DIU Metric

We propose a new metric that describes the Discrepancy between Intersection and Union (DIU) as a strict measure for topological accuracy. The metric is based on the linear map $i_* : H_*(F_\epsilon(P) \cap F_{2\epsilon}(G)) \to H_*(F_\epsilon(P) \cup F_{2\epsilon}(G))$ in homology (with coefficients in the field with two elements, $\mathbb{F}_2$) induced by the inclusion $i : F_\epsilon(P) \cap F_{2\epsilon}(G)) \to F_\epsilon(P) \cup F_{2\epsilon}(G)$ of the intersection into the union. Formally, $\xi^{err}$ is defined as

$$\xi^{err} = \dim(\ker i_*) + \dim(\operatorname{coker} i_*). \quad (4)$$

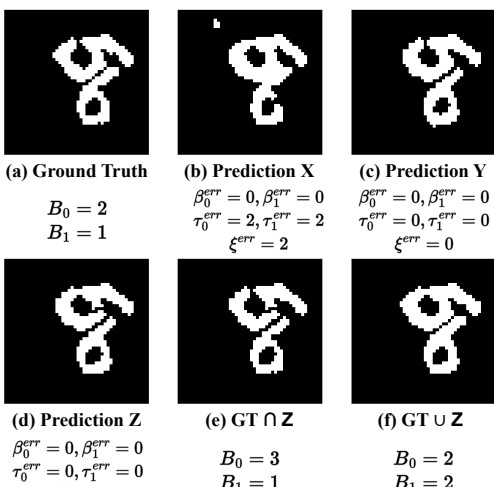

**(a) Ground Truth**
$B_0 = 2$
$B_1 = 1$

**(b) Prediction X**
$\beta_0^{err} = 0, \beta_1^{err} = 0$
$\tau_0^{err} = 2, \tau_1^{err} = 2$
$\xi^{err} = 2$

**(c) Prediction Y**
$\beta_0^{err} = 0, \beta_1^{err} = 0$
$\tau_0^{err} = 0, \tau_1^{err} = 0$
$\xi^{err} = 0$

**(d) Prediction Z**
$\beta_0^{err} = 0, \beta_1^{err} = 0$
$\tau_0^{err} = 0, \tau_1^{err} = 0$
$\xi^{err} = 2$

**(e) GT ∩ Z**
$B_0 = 3$
$B_1 = 1$

**(f) GT ∪ Z**
$B_0 = 2$
$B_1 = 2$

Figure 4: Topological metrics for the characterization of different network predictions (b-d) with a given ground truth (a). Evaluating Betti numbers does not favor Pred. X over Y. Similarly, the Betti matching metric does not favor Y over Z. Only the DIU metric (comparing intersection (e) and union (f)) prefers the semantically favorable Y over Z.

By Alexander duality, this quantity can be expressed purely in terms of connected components (homology in degree 0) of foreground and background. Writing $j : B_\epsilon(P) \cap B_{2\epsilon}(G) \to B_\epsilon(P) \cup B_{2\epsilon}(G)$ of the intersection of backgrounds into their union. We have

$$\dim(\ker i_1) = \dim(\operatorname{coker} j_0), \text{ and } \dim(\operatorname{coker} i_1) = \dim(\ker j_0). \tag{5}$$

Thus, we have

$$\xi^{err} = \dim(\ker i_0) + \dim(\operatorname{coker} i_0) + \dim(\ker j_0) + \dim(\operatorname{coker} j_0). \tag{6}$$

Intuitively, the DIU metric $\xi^{err}$ counts the number of components in the union that do not have a counterpart in the intersection ($\dim(\operatorname{coker})$) and the surplus of intersection components that correspond to the same component in the union ($\dim(\ker)$). Figure 4 (d) shows an example of cases where the Betti number error and the Betti matching error both fail to capture the semantic difference between ground truth and prediction. We provide more examples of DIU metric scores in the supplementary material Figures 11 and 12. The DIU metric is similar to counting *false splits* and *false merges* (e.g., (Funke et al., 2017)) as it also counts topological errors in dimension 0. However, the DIU considers topological differences between union and intersection and errors in dimension 1. See Suppl. A.9 and Berger et al. (2024a) for a detailed comparison.

## 2.6 COMPUTATIONAL COMPLEXITY

Compared to 2D PH approaches ($\mathcal{O}(n \log n)$ (Edelsbrunner & Harer, 2022)), our method's asymptotic complexity is lower. The creation of $\mathcal{G}(P, G)$ using connected component labeling (Wu et al., 2009) and extraction of the node labels can be done in $O(n \cdot \alpha(n))$, where $\alpha$ is the inverse Ackermann function, upper-bounded by 5 in practice. Identifying the *regular nodes* depends on evaluating the node's 1-hop neighborhood. Given the graph's planarity, the number of edges is upper bounded by $O(n)$. Therefore, evaluating the direct neighborhood of all nodes is possible in linear time. For the final aggregation of pixel scores, every pixel is evaluated at most once, which preserves the linear complexity. In summary, our loss $\mathcal{L}_{CG}$ can be computed in linear time $O(n \cdot \alpha(n))$. Empirical evaluations on the runtime are provided in the results section; see Figure 6 and Table 2.

## 2.7 ADAPTABILITY

Our method provides a foundation for an efficient identification of topologically critical regions. Within this flexible framework, many adaptations to the method are possible. In the following, we introduce two adaptations that we found beneficial for the performance in specific tasks.

**Aggregation Mode** Our method allows for flexible adaptation of the aggregation mode to fit task-specific needs. We show ablations on this hyperparameter in Tables 5, 6, and 7. Figure 5 shows a comparison of the dense support for the gradient that is achieved with a mean aggregation compared to the support of PH-based methods. More specialized methods can be easily implemented beyond the simple aggregations we evaluate in the ablation. Aggregations, including specific quantiles of the pixels, or aggregating distinguished points such as local maxima or saddle points, are possibilities.

**Threshold Variation Parameter** Finally, we introduce a threshold variation parameter. This parameter $\sigma$ defines the scale of a Gaussian distribution with location $\mu = 0$. This distribution is used to sample a shift parameter $x_{sh}$ to randomly alter the binarization threshold $bin_{th} = 0.5 + x_{sh}$. Introducing this parameter strongly enhanced our method and mitigates information loss due to the binarization step. We provide an ablation on the binarization threshold in Table 3.

## 3 EXPERIMENTS

**Datasets** We evaluate our method on three real-world binary and two multi-class segmentation tasks using publicly available datasets (binary: Cremi (Funke et al., 2018), Roads (Mnih, 2013), Buildings (Mnih, 2013); multi-class: Platelet (Guay et al., 2021), TopCoW (Yang et al., 2023)). We selected tasks from different modalities with different image sizes where topological correctness represents an important characteristic, particularly in the context of downstream applications. Please refer to the supplement for more details on the used datasets. Following Berger et al. (2024b), we projected the 3D volume in the TopCoW dataset to obtain 2D images and used 2D slices for the

Figure 5: Visualization of the pixels that support the gradient of different losses for an exemplary label-prediction pair. The support pixels are displayed as a white overlay over the prediction. The gradient of pixel-wise loss functions (e.g. CE) is supported by every incorrectly predicted pixel. The BM gradient is supported by two pixels for every topological feature. The Topograph (ours) gradient is supported by every pixel in the incorrectly predicted and topologically relevant regions.

Cremi and Platelet datasets. To apply the topological losses in the multiclass setting, we frame the multiclass problem as multiple binary classification problems as proposed in (Berger et al., 2024b). This approach scales the computational complexity linearly with the number of classes compared to a single binary loss calculation of the respective loss function.

**Baselines**   In all of our experiments, we compare our method to **Dice loss**. This is important to validate that our loss (1) does not significantly impair pixel-wise accuracy and (2) truly improves topological correctness. Moreover, we compare our method to the task-specific **clDice** method (Shit et al., 2021) that is especially well-suited for the tubular structured roads, Cremi, and circle of Willis datasets. Next, we compare to the Mosin loss function (Mosinska et al., 2018), which uses a VGG's feature representation for loss calculation. Finally, we compare to PH-based approaches. First, **HuTopo** proposed by (Hu et al., 2019), which maximizes the similarity of the predictions persistence diagram to a corresponding ground truth diagram. Second, the refined **Betti matching loss**, which further matches the barcodes in the persistence diagrams based on spatial correspondence.

**Evaluation metrics**   We evaluate the **pixel-wise accuracy** using the **Dice score**. To evaluate topological correctness, we report the **clDice** metric (Shit et al., 2021), the **Betti number error (B0, B1)** in dimensions 0 and 1, and the more refined **Betti matching error (BM)** (Stucki et al., 2023), which considers the spatial alignment of topological features in both dimensions. Furthermore, we evaluate the **DIU** metric, which additionally measures topological similarity between union and intersection of label and prediction pairs (see Section 2.5).

**Training and model selection**   We train a U-Net architecture (Ronneberger et al., 2015) with residual units from scratch and use the Adam (Kingma, 2014) optimizer. We perform 5-fold cross-validation and evaluate on an independent test set. We perform a random hyperparameter search with 25 runs on each split and select the model with the highest balanced Dice and BettiMatching score on the validation set. We report the mean performance and standard deviation on the independent test sets across the five data splits. Please refer to the supplement for specifics on the hyperparameter spaces and more details on the training. We use the paired t-test between our model and each baseline to evaluate statistically significant performance ($p$-value $< 0.05$) improvements.

## 3.1   RESULTS

**Binary results**   Our method exhibits improved topological accuracy, as shown by the best values on the DIU and BM metrics, with significant improvements compared to most baselines on the Roads dataset. Compared to loss functions such as HuTopo and Dice, we achieve significant performance improvements. Interestingly, the HuTopo baseline shows a very low B0 score and a high BM and DIU error. This observation indicates that HuTopo optimizes for the correct number of topological features but disregards spatial alignment, which is in agreement with its theoretical limitation of spatial mismatches (Stucki et al., 2023). Furthermore, we observe that the Dice loss yields a (naturally) high Dice and clDice score. Although this difference is insignificant, it comes at the cost of reduced topological correctness across all topological metrics. On the Roads dataset, we see that Betti Matching also achieves good DIU and BM scores, almost as low as ours, but performs worse in clDice. Specifically, the effects of a homotopy equivalence between union and intersection (achieved when minimizing our loss function, as described in Section 2.3) are especially pronounced for tubular structures (such as roads) and also typically lead to a low clDice score, as a mutual inclusion of the centerlines also implies this homotopy equivalence (see Suppl. Figure 11). For the Cremi

Table 1: Quantitative results. Best performances indicated in **bold**, statistical significance underlined ($p$-value $< 0.05$). Our method outperforms the baselines in strict topological metrics (DIU, BM) for all but one datasets while achieving similar Dice.

| Dataset | Loss | DIU ↓ | BM ↓ | B0 ↓ | B1 ↓ | Dice ↑ | clDice ↑ |
|---|---|---|---|---|---|---|---|
| Buildings | Dice | $79.798_{\pm 4.70}$ | $67.101_{\pm 2.80}$ | $25.867_{\pm 2.51}$ | $1.452_{\pm .12}$ | $\mathbf{.777}_{\pm .01}$ | $.611_{\pm .02}$ |
| | ClDice | $80.867_{\pm 4.48}$ | $64.771_{\pm 2.37}$ | $\underline{26.188}_{\pm 1.35}$ | $\mathbf{1.407}_{\pm .06}$ | $.774_{\pm .01}$ | $\mathbf{.623}_{\pm .01}$ |
| | HuTopo | $\underline{86.155}_{\pm 4.52}$ | $69.662_{\pm 3.36}$ | $\mathbf{16.062}_{\pm 5.50}$ | $1.481_{\pm .07}$ | $\underline{.764}_{\pm .01}$ | $.583_{\pm .01}$ |
| | BettiM. | $79.271_{\pm 4.24}$ | $63.310_{\pm 2.62}$ | $17.669_{\pm 2.35}$ | $\mathbf{1.407}_{\pm .10}$ | $.774_{\pm .02}$ | $.605_{\pm .03}$ |
| | Mosin | $81.724_{\pm 4.90}$ | $\underline{68.875}_{\pm 5.15}$ | $\underline{29.157}_{\pm 4.83}$ | $1.495_{\pm .08}$ | $\underline{.763}_{\pm .01}$ | $.581_{\pm .02}$ |
| | Ours | $\mathbf{77.824}_{\pm 3.48}$ | $\mathbf{62.652}_{\pm 3.01}$ | $20.281_{\pm 1.27}$ | $1.443_{\pm .06}$ | $.769_{\pm .01}$ | $.590_{\pm .02}$ |
| Roads | Dice | $7.313_{\pm .61}$ | $\underline{6.615}_{\pm .53}$ | $\underline{1.935}_{\pm .42}$ | $2.642_{\pm .17}$ | $\mathbf{.819}_{\pm .01}$ | $\mathbf{.720}_{\pm .01}$ |
| | ClDice | $\underline{7.127}_{\pm .32}$ | $\underline{5.810}_{\pm .29}$ | $\underline{1.515}_{\pm .21}$ | $3.058_{\pm .14}$ | $\underline{.803}_{\pm .01}$ | $.704_{\pm .01}$ |
| | HuTopo | $\underline{7.498}_{\pm .74}$ | $\underline{6.356}_{\pm .67}$ | $1.185_{\pm .62}$ | $2.683_{\pm .38}$ | $.817_{\pm .01}$ | $.714_{\pm .01}$ |
| | BettiM. | $6.733_{\pm .36}$ | $5.908_{\pm .23}$ | $\mathbf{0.942}_{\pm .10}$ | $\mathbf{2.317}_{\pm .08}$ | $.818_{\pm .01}$ | $.707_{\pm .01}$ |
| | Mosin | $7.221_{\pm .71}$ | $\underline{6.352}_{\pm .64}$ | $1.523_{\pm .38}$ | $2.756_{\pm .11}$ | $.816_{\pm .01}$ | $.710_{\pm .01}$ |
| | Ours | $\mathbf{6.521}_{\pm .47}$ | $\mathbf{5.635}_{\pm .40}$ | $1.212_{\pm .25}$ | $2.619_{\pm .16}$ | $.817_{\pm .01}$ | $.711_{\pm .01}$ |
| Cremi | Dice | $21.840_{\pm .12}$ | $\underline{10.872}_{\pm .27}$ | $1.264_{\pm .12}$ | $2.936_{\pm .09}$ | $.946_{\pm .01}$ | $.960_{\pm .01}$ |
| | ClDice | $21.728_{\pm .35}$ | $\mathbf{10.368}_{\pm .13}$ | $1.296_{\pm .06}$ | $2.848_{\pm .18}$ | $\underline{.944}_{\pm .01}$ | $\mathbf{.961}_{\pm .01}$ |
| | HuTopo | $23.192_{\pm .72}$ | $\underline{12.232}_{\pm .52}$ | $1.368_{\pm .04}$ | $2.768_{\pm .17}$ | $\underline{.942}_{\pm .01}$ | $.956_{\pm .01}$ |
| | BettiM. | $22.832_{\pm .56}$ | $10.880_{\pm .38}$ | $\mathbf{0.920}_{\pm .06}$ | $3.608_{\pm .33}$ | $\underline{.927}_{\pm .01}$ | $.950_{\pm .01}$ |
| | Mosin | $21.984_{\pm .37}$ | $10.944_{\pm .20}$ | $1.312_{\pm .07}$ | $\mathbf{2.704}_{\pm .17}$ | $.946_{\pm .01}$ | $.960_{\pm .01}$ |
| | Ours | $\mathbf{21.272}_{\pm .29}$ | $10.392_{\pm .22}$ | $1.224_{\pm .09}$ | $2.720_{\pm .14}$ | $\mathbf{.947}_{\pm .01}$ | $\mathbf{.961}_{\pm .01}$ |
| Platelet | Dice | $\underline{14.791}_{\pm 1.41}$ | $1.610_{\pm 0.19}$ | $0.640_{\pm 0.10}$ | $0.536_{\pm 0.07}$ | $\mathbf{.752}_{\pm .01}$ | $.822_{\pm .01}$ |
| | ClDice | $\underline{50.156}_{\pm 29.09}$ | $6.407_{\pm 4.08}$ | $2.949_{\pm 2.02}$ | $3.028_{\pm 2.07}$ | $.728_{\pm .01}$ | $.783_{\pm .02}$ |
| | HuTopo | $\underline{11.523}_{\pm 0.36}$ | $1.119_{\pm 0.05}$ | $0.372_{\pm 0.03}$ | $\mathbf{0.376}_{\pm 0.01}$ | $.746_{\pm .01}$ | $.823_{\pm .00}$ |
| | BettiM. | $13.000_{\pm 1.32}$ | $1.239_{\pm 0.09}$ | $0.400_{\pm 0.03}$ | $0.451_{\pm 0.06}$ | $.747_{\pm .01}$ | $.826_{\pm .01}$ |
| | Mosin | $11.143_{\pm 0.52}$ | $1.130_{\pm 0.06}$ | $\mathbf{0.356}_{\pm 0.05}$ | $\underline{0.407}_{\pm 0.02}$ | $.747_{\pm .01}$ | $\underline{.835}_{\pm .00}$ |
| | Ours | $\mathbf{10.906}_{\pm 0.26}$ | $\mathbf{1.110}_{\pm 0.04}$ | $0.406_{\pm 0.02}$ | $\mathbf{0.376}_{\pm 0.02}$ | $.751_{\pm .01}$ | $\mathbf{.843}_{\pm .00}$ |
| TopCoW | Dice | $15.716_{\pm 1.61}$ | $0.977_{\pm 0.89}$ | $0.722_{\pm 0.07}$ | $0.073_{\pm 0.02}$ | $.729_{\pm .01}$ | $.773_{\pm .01}$ |
| | ClDice | $10.670_{\pm 1.76}$ | $0.678_{\pm 0.13}$ | $0.483_{\pm 0.11}$ | $\mathbf{0.049}_{\pm 0.01}$ | $.733_{\pm .01}$ | $\mathbf{.804}_{\pm .02}$ |
| | HuTopo | $16.057_{\pm 6.67}$ | $0.992_{\pm 0.43}$ | $0.717_{\pm 0.33}$ | $0.092_{\pm 0.05}$ | $.711_{\pm .04}$ | $\underline{.758}_{\pm .04}$ |
| | BettiM. | $12.352_{\pm 0.90}$ | $0.761_{\pm 0.06}$ | $0.556_{\pm 0.06}$ | $0.064_{\pm 0.01}$ | $\mathbf{.740}_{\pm .01}$ | $\underline{.787}_{\pm .02}$ |
| | Mosin | $23.534_{\pm 16.95}$ | $1.489_{\pm 0.98}$ | $1.128_{\pm 0.83}$ | $\underline{0.154}_{\pm 0.07}$ | $.606_{\pm .16}$ | $.659_{\pm .17}$ |
| | Ours | $\mathbf{10.477}_{\pm 1.35}$ | $\mathbf{0.658}_{\pm 0.09}$ | $\mathbf{0.461}_{\pm 0.06}$ | $0.052_{\pm 0.02}$ | $.735_{\pm .01}$ | $.801_{\pm .01}$ |

dataset, our methods consistently exhibits strong performance across all metrics. clDice performs comparably strong, since the tubular structure of the data is beneficial for the centerline-based loss calculation. Some of the other methods have a strong performance in one metric but worse results on other metrics.

**Multi-class results** In both multiclass datasets, our method consistently surpasses all baselines in the two key topological metrics, DIU and BM score, often showing a statistically significant performance improvement. For the B0 and B1 errors, our method performs comparably or slightly below the top-performing baseline (e.g., HuTopo). However, our method's superiority in BM score indicates that this is likely caused by spatial mismatches of topological features, which is also backed by our qualitative results (see Figures 15 and 16). Additionally, we note that our loss formulation does not compromise pixel-wise accuracy, as reflected in the Dice score, where our method matches the performance of the best baseline across both datasets, even significantly outperforming clDice and HuTopo on the Platelet dataset.

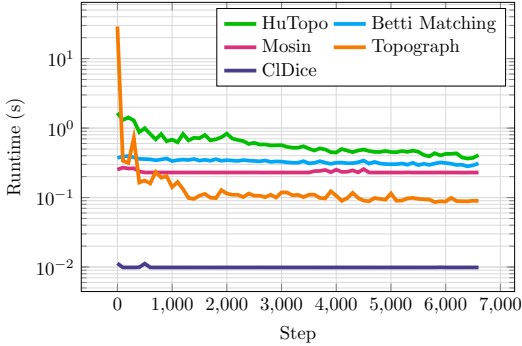

Figure 6: Runtime for a single loss calculation over the complete training process.

In the TopCoW dataset, which is a vessel dataset, clDice emerges as the strongest baseline. However, its specialization in tubular structures and reliance on extracting accurate centerlines leads to poor performance in the Platelet dataset, where blob-like structures with inclusions dominate. We also observe that, on the TopCoW dataset, Betti Matching yields a relatively good BM score while having a significantly worse DIU score. This observation is likely caused by Betti Matching's weaker theoretical guarantees that do not account for homotopy equivalence between intersection and union. This difference is further underlined by Betti Matching's significantly lower clDice score, similar to our binary experiments. In the Platelet dataset, the HuTopo baseline performs strongly, a notable contrast to its low performance on the TopCoW dataset and in binary experiments, where spatial mismatches in topological structures become more frequent.

**Ablation on runtime** We empirically measure the runtime of our loss compared to other topological losses. Figure 6 shows the time demand of different loss functions across an exemplary training run on the Platelet dataset. After a short run-in phase, our method is consistently faster than PH-based approaches and the Mosin baseline. This is also reflected in the accumulated training times in Table 2, which shows that our method saves up to one hour of training time for a single run on the Platelet dataset. For the average loss calculation, we achieve a 3-6 fold improvement compared to the PH methods.

Table 2: Computational runtime.

| Loss | Avg. loss calc. | Train time |
|---|---|---|
| ClDice | 9.88 ms | 1h27m |
| Mosin | 233.00 ms | 2h29m |
| Hu et al. | 602.99 ms | 3h05m |
| BettiM. | 327.64 ms | 2h46m |
| Ours | 95.94 ms | 1h53m |

## 3.2 Ablation on the binarization threshold variation

We investigate the effect of the binarization variation parameter, described in Section 2.7 in Table 3 The results indicate that a fixed binarization without variation is inferior to additional Gaussian threshold variation. A standard deviation of 0.05 - 0.1 yields the best results.

## 4 Conclusion

This work proposes a novel framework for image segmentation to identify topologically critical regions via a component graph $\mathcal{G}(P, G)$. Our proposed loss function improves topological accuracy over pixel-wise losses, consistently delivers state-of-the-art performance compared to other topology-preserving approaches, and is low in runtime. Formally, Topograph provides stricter topological guarantees than existing methods. Our method goes beyond ensuring homotopy equivalence between ground truth and segmentation. It further enforces homotopy equivalence to both their union and intersection through the respective inclusion maps, thereby capturing the spatial correspondence of their topological properties. Additionally, we propose a sensitive segmentation metric (DIU) capturing fine topological discrepancies which cannot be univocally captured by existing metrics.

Table 3: Ablation on the binarization threshold variation for the buildings dataset. Best results are in **bold** and second best in *italics*.

| Thres. Var. | DIU ↓ | BM ↓ | Dice ↑ |
|---|---|---|---|
| 0. | 51.5625 | 40.6250 | 0.80886 |
| 0.01 | 47.2500 | 37.7500 | 0.81965 |
| 0.05 | **46.1250** | *36.2500* | 0.79447 |
| 0.1 | *46.7500* | **35.5625** | **0.82555** |
| 0.2 | 47.8750 | 38.2500 | 0.80861 |
| 0.5 | 49.2500 | 36.5625 | *0.82431* |
| Otsu | 49.1250 | 37.0000 | 0.82396 |

**Limitations and future work** PH-based methods naturally define a filtration on pixel intensities, capturing topological information across all thresholds. While binarization trades off some of this information, it significantly enhances runtime efficiency. Despite this, in many scenarios, our method surpasses PH-based methods in segmentation performance. Ablation studies on the threshold variation parameter highlight the value of including topological information beyond a fixed threshold of 0.5, with features near the binarization threshold proving particularly important in our experiments (see Table 3). Future efforts should aim to explore how our method, with its strict topological guarantees, can be integrated with filtrations to capture topological information at all thresholds, further enhancing performance in every iteration. Additionally, our framework is currently designed for 2D images, and extending the method to 3D remains a promising direction for future work.

## 5 REPRODUCIBILITY

We are committed to making our work entirely reproducible for the scientific community. Our source code, released under an open-source license, is available via an anonymous public GitHub repository `https://github.com/AlexanderHBerger/Topograph`. We provide all checkpoints for all the models trained with our method and all the baselines and make them available upon request. An exemplary model checkpoint is available in the repository. All datasets used in the experiments are publicly available and are described in detail in the Supplementary Material. All applied preprocessing is documented in the code on github. We provide details on our random and equal hyperparameter search on each data-split and baseline in the main manuscript and supplement (see Table 9). We provide an extensive schematic and intuitive description of our method and proofs in the Supplementary Material. Moreover, we provide numerous qualitative examples to help the understanding of our newly presented DIU metric in Figures 11 and 12. Lastly, we provide further qualitative examples of the combined graph in Figure 10.

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

# A APPENDIX

## A.1 SCHEMATIC TOPOGRAPH LOSS

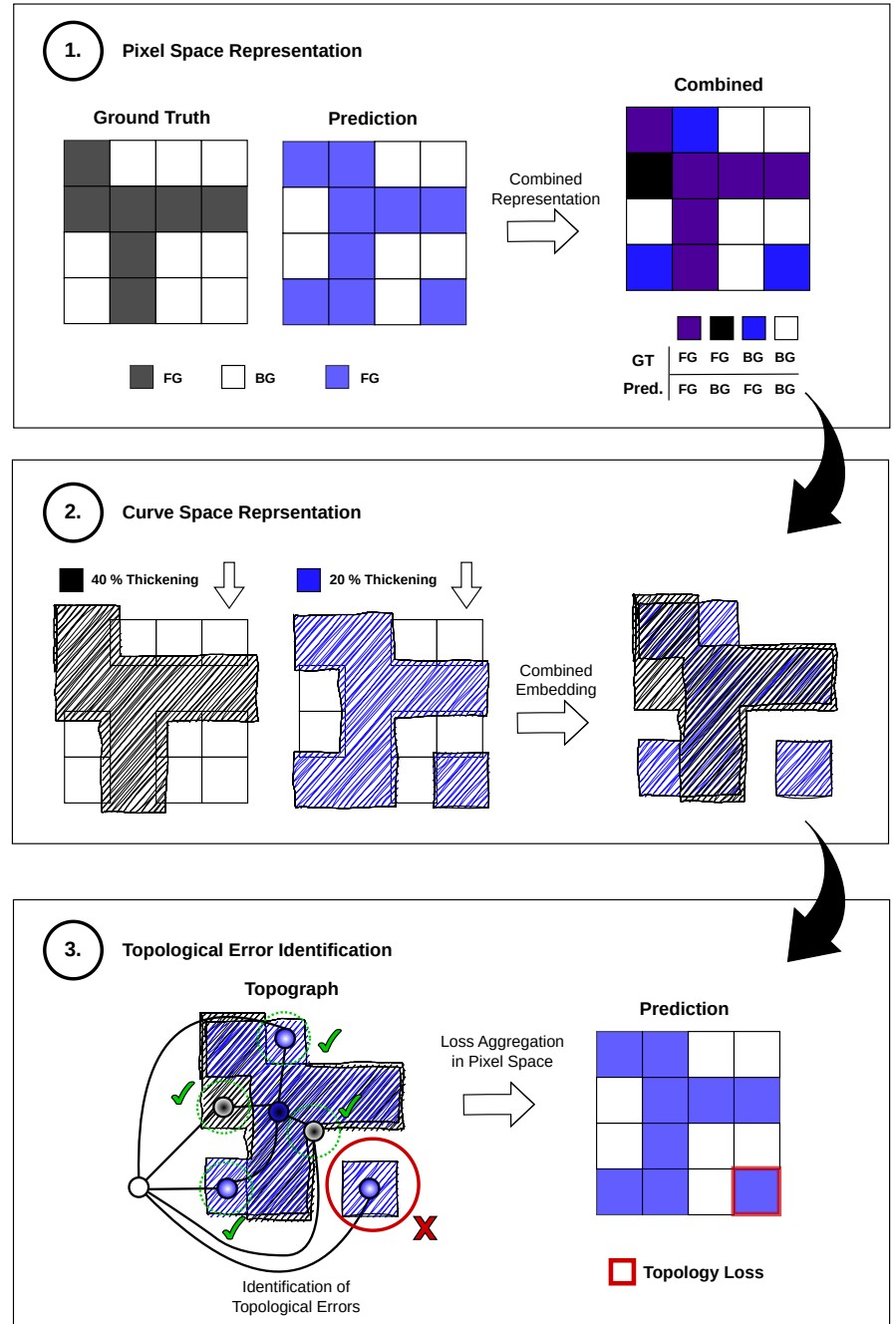

Figure 7: Overview of the loss calculation using our method. (1.) First, the information of prediction and ground truth is combined. Pixels with a correct prediction (purple (FG, FG), white (BG, BG)) do not influence the loss calculation. (2.) To guarantee that the prediction and ground truth foreground areas only have transversal cuts, the areas are thickened with different margins. (3) Based on a region adjacency graph of the topologically critical areas can be identified and related to the relevant pixels.

## A.2 ZERO LOSS PROOF

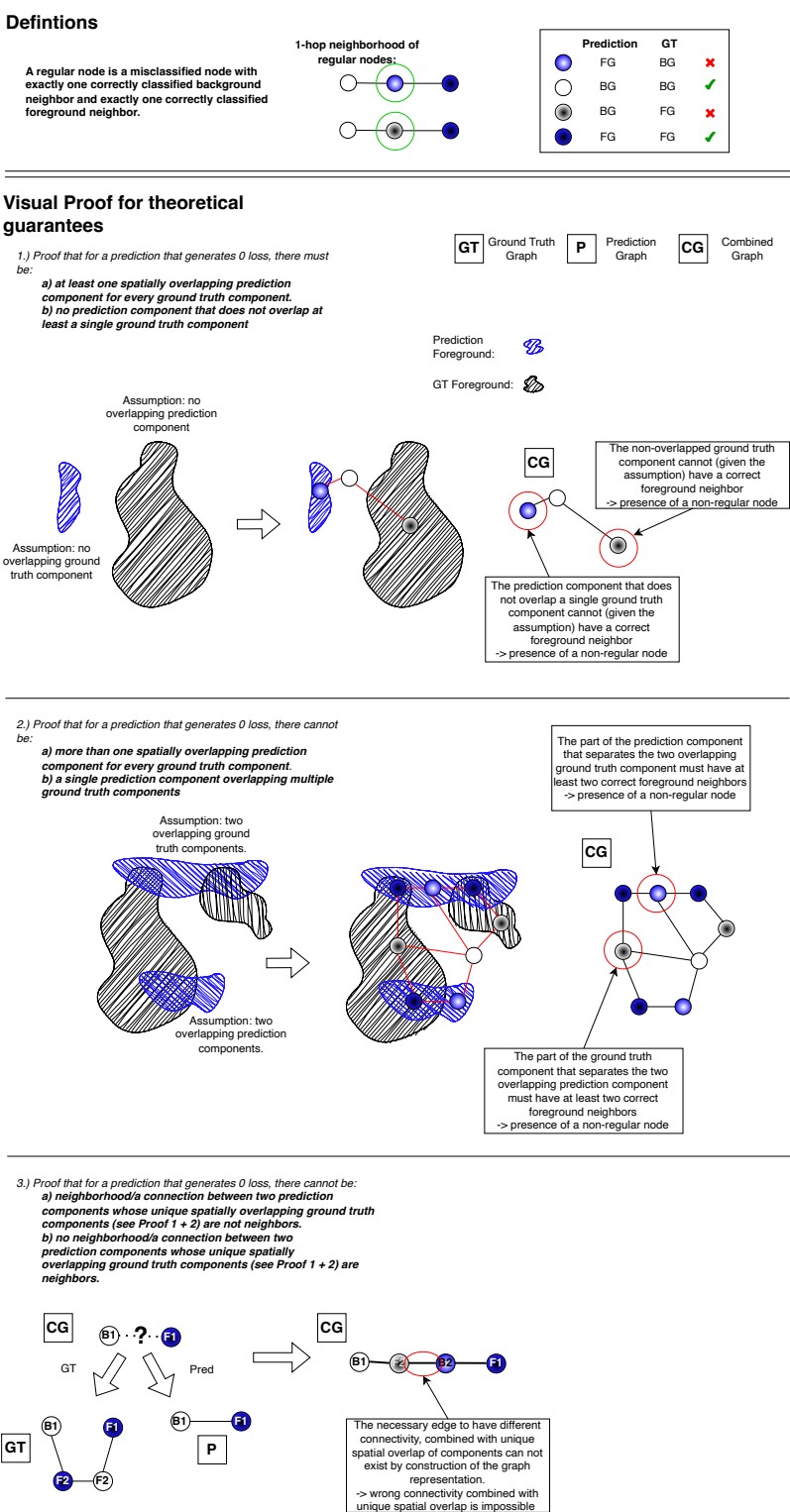

Figure 8: Visual proof that a zero loss guarantees topological equivalence between ground truth and prediction. First, we present our definition of regular nodes. In the following, we prove topological equivalence in three steps using contradiction.

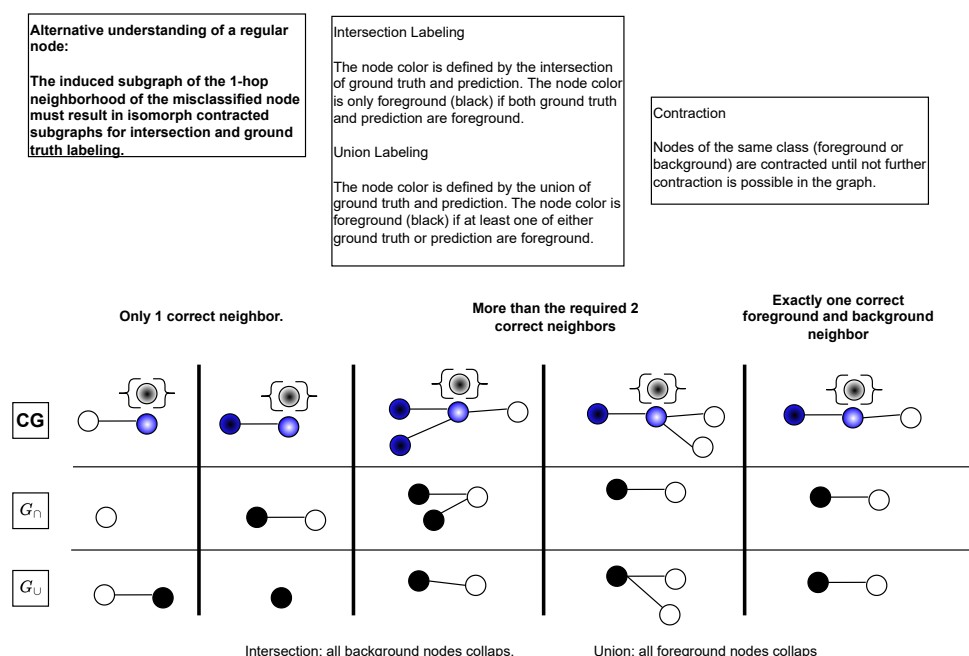

Figure 9: Visualization of the properties of a regular node. Only a regular node results in isomorphic contracted subgraphs after intersection and union labeling.

Figure 9 describes how errors with different neighborhoods behave under labeling according to the intersection or union. Regular nodes (exactly one correct foreground and background neighbor) result in the same graph independent of intersection and union. It follows that these nodes do not affect the isomorphism of the intersection and the union graph.

### A.3 ADDITIONAL GRAPH EXAMPLES

Fig. 10 provides additional examples of the combined graph construction and the difference between critical and regular nodes.

## Dimension 0 Matching Cases:

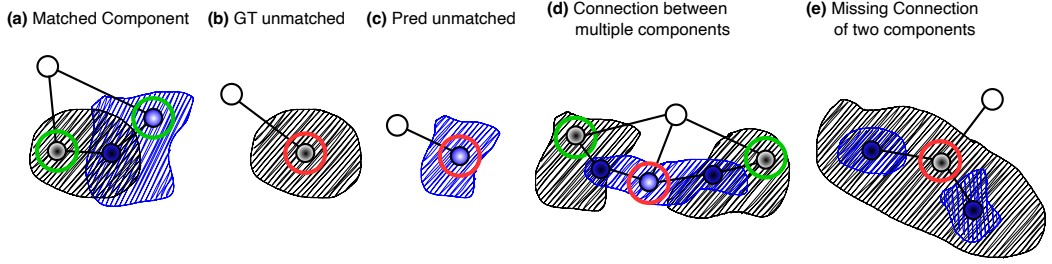

## Dimension 1 Matching Cases:

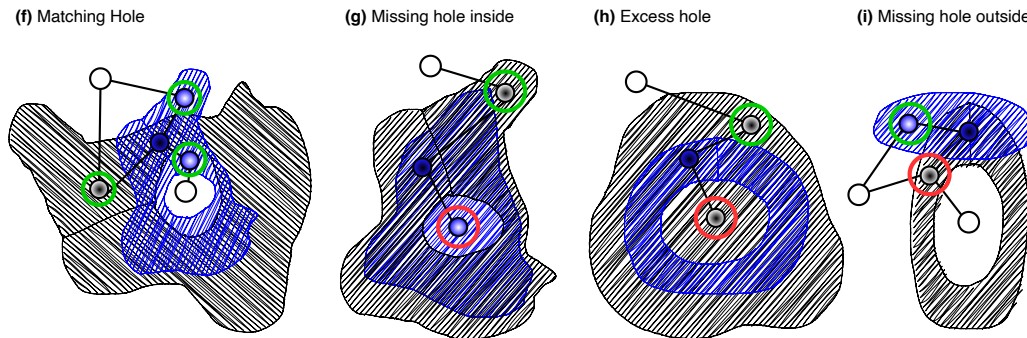

## Union/Intersection errors:

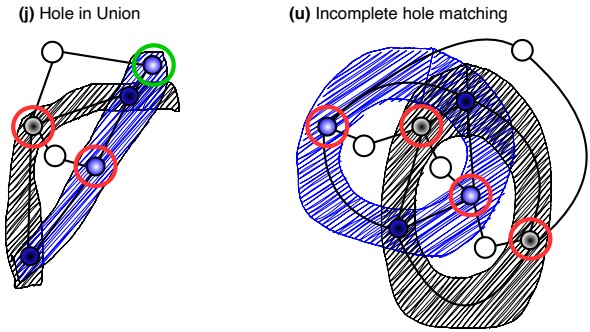

Figure 10: Additional examples of our combined graph representation with critical nodes marked by a red circle and regular nodes marked by a green circle. Correctly predicted nodes are not marked. The combined graph is visualized as an overlay following the notation from Fig. 8. The three rows focus on frequent topological structures in Dimension 0 and 1 as well as two examples for Union/Intersection errors.

### A.4 DIU EXAMPLES

Fig. 11 visualizes the difference between multiple topological performance metrics and the practical implication of homotopy equivalence between union and intersection in the case of a vessel example. Fig. 12 shows comparisons between different performance metrics on further examples.

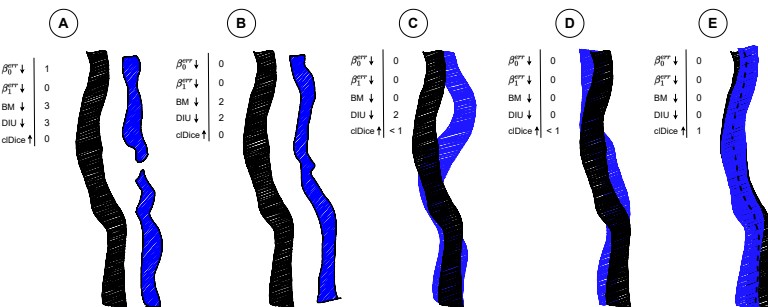

Figure 11: Comparison of different topological performance metrics on the example of tubular structure segmentation. (A) results in a Betti number (+ all stricter metrics) error since the disconnected blue vessel builds two connected components. (B) results in a Betti matching error (+ all stricter metrics) because the spatial correspondence between the blue and black vessels is not provided. (C) results in and DIU error because the two vessels lose the spatial correspondence in parts and then connect again. (D) results in a clDice score $< 1$ because the overlap between the vessels does not include the skeleta in all parts.

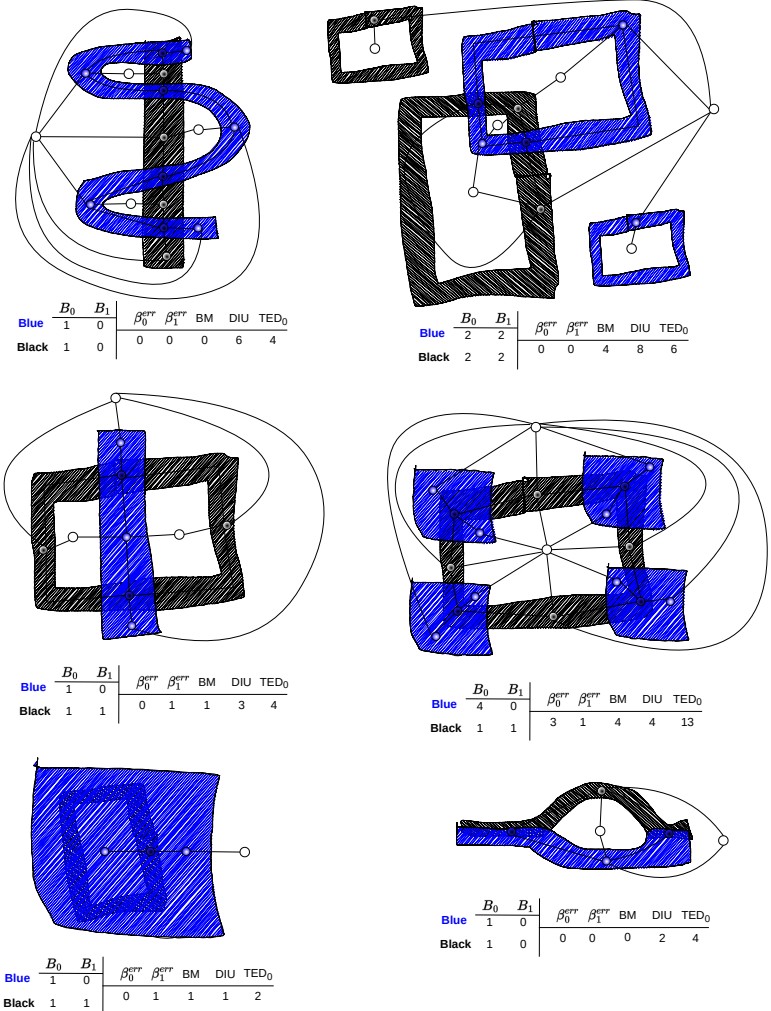

Figure 12: Examples for comparing topological performance metrics with different strictness. In the top left and bottom right examples only the DIU metric captures a topological error. The combined graph is visualized as an overlay following the notation from Fig. 8. We further compare to the TED metric Funke et al. (2017) with 0 boundary tolerance. For TED the background is assumed to be a single instances with potentially disconnected parts. Foreground components are viewed as independent instances with separate labels.

## A.5 ABLATION ON ALPHA PARAMETER

We study the effect of the loss weighting via an alpha parameter in an ablation experiment. The results, see Table 4, show that there is an optimal weighting of the parameter in combination to a pixel-wise loss.

Table 4: Ablation on the alpha parameter. The provided results are validation scores on the TopCoW dataset. Rows where the best performance was achieved before the activation of the topological loss are marked with a star *. The best results are in **bold**, and the second best results are in *italics*.

| Alpha | DIU ↓ | BM ↓ | B0 ↓ | B1 ↓ | Dice ↑ | clDice ↑ |
|---|---|---|---|---|---|---|
| 0.0 | 10.944 | 0.7148 | 0.5259 | 0.0704 | 0.7461 | 0.7912 |
| 0.1 | 7.1667 | 0.4815 | 0.3296 | **0.0407** | 0.7540 | 0.8238 |
| 0.2 | **6.5556** | **0.4407** | **0.3037** | **0.0407** | 0.7571 | 0.8279 |
| 0.3 | 7.6111 | 0.4926 | 0.3259 | *0.0556* | *0.7601* | *0.8293* |
| 0.4 | 7.7222 | 0.4815 | 0.3370 | *0.0556* | **0.7770** | **0.8453** |
| 0.5* | 11.3889 | 0.7148 | 0.5074 | 0.0667 | 0.7478 | 0.8047 |
| 0.6 | *7.0556* | *0.4630* | *0.3074* | 0.0815 | 0.7372 | 0.8032 |
| 0.7* | 11.3889 | 0.7148 | 0.5074 | 0.0667 | 0.7478 | 0.8047 |
| 0.8* | 11.3889 | 0.7148 | 0.5074 | 0.0667 | 0.7478 | 0.8047 |
| 0.9* | 11.3889 | 0.7148 | 0.5074 | 0.0667 | 0.7478 | 0.8047 |
| 1.0* | 11.3889 | 0.7148 | 0.5074 | 0.0667 | 0.7478 | 0.8047 |

## A.6 ABLATION ON THE AGGREGATION

The results of the ablations on the aggregation mode are shown in Tables 6 and 5. The results show that the aggregation mode is an influential hyperparameter with varying optima depending on the dataset. For the roads dataset, dense aggregations such as mean and root mean square (rms) that include information of all pixels in the critical region perform better than the sparse max aggregation. In the buildings dataset root mean square provides a compromise of high pixel-wise accuracy combined with good topological correctness.

Table 5: Ablation on the aggregation type for pixels within incorrect nodes. The provided results are validation scores on the roads dataset. Best results are in **bold** and second best results are in *italics*.

| Aggregation | DIU ↓ | BM ↓ | B0 ↓ | B1 ↓ | Dice ↑ | clDice ↑ |
|---|---|---|---|---|---|---|
| Mean | 4.70 | *3.95* | **0.40** | *1.75* | *0.90264* | **0.84036** |
| Max | *4.50* | 4.05 | **0.40** | 2.05 | 0.88350 | 0.83857 |
| RMS | **4.25** | **3.90** | *0.45* | **1.65** | 0.90137 | 0.83668 |
| Sum | 5.90 | 5.76 | 1.50 | 2.05 | 0.90140 | 0.83870 |
| CE | 4.80 | 4.00 | 0.75 | *1.75* | **0.90320** | *0.83908* |

Table 6: Ablation on the aggregation type for pixels within incorrect nodes. The provided results are validation scores on the buildings dataset. Best results are in **bold** and second best results are in *italics*.

| Aggregation | DIU ↓ | BM ↓ | B0 ↓ | B1 ↓ | Dice ↑ | clDice ↑ |
|---|---|---|---|---|---|---|
| Mean | *47.7500* | *37.7500* | 11.0000 | **0.3750** | 0.79831 | 0.66169 |
| Max | **45.7500** | **37.3125** | 12.1250 | 0.5625 | 0.81180 | 0.69061 |
| RMS | 48.8125 | 38.1250 | **9.1875** | 0.8125 | **0.82333** | *0.70351* |
| Sum | 49.3125 | 39.4375 | 12.2500 | *0.4375* | 0.81112 | 0.68048 |
| CE | 51.6875 | 39.0625 | *9.3125* | 0.5000 | *0.81459* | **0.70370** |

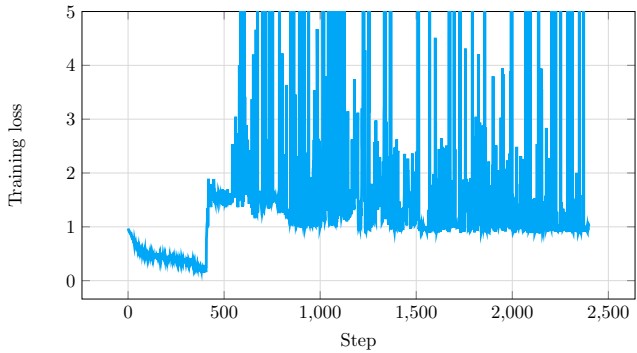

Figure 13: Training loss when setting $\alpha$ to a very high value. After the warmup phase (here around step 400), the topograph loss dominates the training, leading to deteriorating performance.

Table 7: Ablation on the aggregation type for pixels within incorrect nodes. The provided results are validation scores on the platelet dataset. Best results are in **bold** and second best results are in *italics*.

| Aggregation | DIU ↓ | BM ↓ | B0 ↓ | B1 ↓ | Dice ↑ | clDice ↑ |
|---|---|---|---|---|---|---|
| Mean | *11.25722* | *1.18968* | *0.31548* | 0.54008 | *0.78641* | 0.86118 |
| Max | **10.86111** | **1.1496** | **0.29603** | **0.52024** | 0.77875 | **0.86652** |
| RMS | 11.66111 | 1.22381 | 0.31667 | 0.53413 | 0.77651 | *0.86383* |
| Min | 12.54722 | 1.33413 | 0.38929 | *0.53373* | **0.78872** | 0.85891 |

## A.7 ABLATION ON THE DATASET SIZE

Table 8 shows the results of an ablation study of the dataset size. We found that our method improves performance across all the different dataset sizes on the buildings dataset. Furthermore, the ablation results do not indicate that the relative topological performance improvement gets smaller with an increased amount of data.

Table 8: Comparison of the relative performance increase of using the Topograph method on different dataset sizes. The displayed performances for BM and DIU represent the best validation scores achieved for the respective metrics. The same validation set is used independent of the amount of training data. The ablation was performed on the buildings dataset.

| Dataset Frac. | BM ↓ (TG) | DIU ↓ (TG) | BM ↓ (DSC) | DIU ↓ (DSC) | BM relative improvement ↑ | DIU relative improvement ↑ |
|---|---|---|---|---|---|---|
| 12.5% | 72.8929 | 86.5714 | 76.3333 | 89.1667 | 4.51% | 2.91% |
| 25.0% | 72.5357 | 83.9286 | 74.8095 | 88.3961 | 3.04% | 5.05% |
| 37.5% | 68.5238 | 84.5952 | 75.5238 | 89.6071 | 9.27% | 5.59% |
| 50.0% | 66.7857 | 84.0952 | 67.4881 | 86.1191 | 1.04% | 2.35% |
| 62.5% | 65.2143 | 82.8333 | 65.8095 | 83.3333 | 0.90% | 0.60% |
| 75.0% | 62.9048 | 79.9048 | 64.6191 | 80.0238 | 2.65% | 0.15% |
| 87.5% | 62.2738 | 81.2976 | 65.4881 | 82.0476 | 4.91% | 0.91% |
| 100% | 59.2381 | 76.4167 | 64.3333 | 80.9524 | 7.92% | 5.60% |

## A.8 DETAILED DATASET DESCRIPTION

**Buildings**   In the buildings dataset, the aim is to segment buildings based on aerial images. The topologically interesting aspect is the number of foreground components and also the correct spatial correspondence, ensuring that, e.g., buildings are correctly identified as opposed to parking areas, roads, and other landmarks. We use the buildings dataset created by Mnih (2013). Our training/validation set consists of 80 images with a size of 375x375 and three color channels, the test set contains 21 images of the same size. During training, we used a random crop to a size of 128x128. The validation and test set for this dataset were split into 4 patches of size 128x128 in the 4 quadrants of the image. This results in a total of 64 validation images and 84 test images for every fold.

**Roads**   The task for the roads dataset is to provide a binary segmentation of aerial images into streets and the background. In the roads dataset the correct connectivity is the most interesting aspect. We use the dataset created by Mnih (2013). Our training/validation set consists of 100 images with a size of 375x375 and three color channels, the test set contains 24 images of the same size. During training, we used a random crop to a size of 128x128. The validation and test set for this dataset were split into 4 patches of size 128x128 in the 4 quadrants of the image. This results in a total of 80 validation images and 96 test images for every fold.

**Cremi**   The Cremi dataset Funke et al. (2018) contains electron microscopy images of an adult fly brain. The segmentations consist of many closed circles of which most are connected together and thus form only a few individual connected components. The images are of size 312x312 and have just one gray-scale channel. We use 100 images for training and validation and 25 images as test set. During training we randomly crop an area of 128 x 128 per image. The validation and test set for this dataset were split into 4 patches of size 128x128 in the 4 quadrants of the image. This results in a total of 80 validation images and 100 test images for every fold.

**Platelet**   In this dataset, the aim is to segment round objects where the topology is described by "inclusion," e.g., a mitochondrion is always inside a cell segment. The dataset Guay et al. (2021) contains six different classes (cell, mitochondrion, canalicular channel, alpha granule, dense granule, and dense granule core). The dataset contains 50 samples for training/validation and 25 for testing each (800×800 pixels) with six classes Guay et al. (2021). We create overlapping patches of size 200×200 during our experiments.

**TopCoW**   The goal of the circle of Willis (coW) dataset is the segmentation of the coW and the correct assignment to the 15 different vascular classes. The coW has hypoplastic and absent components across different subjects, making correct segmentation challenging Yang et al. (2023). We project the magentic resonance angiography scans and the labels to a 2D image and segmentation mask. We use the public MICCAI 2023 TopCoW challenge data and use 110 subjects for training/validation and 22 subjects for testing. We crop each image to a size of 100×80 pixels.

## A.9 COMPARISON OF DIU AND BM TO "FALSE SPLITS" AND "FALSE MERGES".

In neuroscience, counting the number of "false splits" and "false merges" is a widespread metric to assess neuron segmentation performance. In this paragraph, we compare the DIU metric to these practices. To make a meaningful comparison, we have to make some prior assumptions. *Definition of "False split/merge":* We define a "false split" by a spatial overlap of two instances in the prediction with a single instance in the ground truth. A "false merge" is then defined by two instances of the ground truth being overlapped by a single instance in the prediction. Moreover, we assume that each instance resembles exactly one connected component with a unique label. To make a comparison to our method, we furthermore assume that there is a background class that shares a common label, but the background can be made up of one or more connected components (but is considered a single instance with a single label). Using the definitions in Funke et al. (2017), the total number of false splits can be calculated by summing the value n-1 for every ground truth instance, where n is the cardinality of the set of spatially overlapped prediction labels. Similarly, the total number of wrong splits can be calculated by summing the value n-1 for every prediction instance, where n is the cardinality of the set of spatially overlapped ground truth labels.

*Background Convention:* Furthermore, we consider two different options for handling the background. First, the background can be viewed as a normal instance and thus contribute equally to the error calculation (A1). Second, the background can be ignored when calculating the error (A2). In the following paragraph, we investigate the effect of these background choices on different error cases.

With the definitions made above and following A1, we now compare how the sum of "false splits" and "false merges" is related to the DIU metric. First, we consider the basic cases of having an instance in the GT and no corresponding instance in the predictions or reversed. Summing up the "false merges" and the "false splits" would result in an error of 1. We can calculate the result as follows: the GT-BG (ground-truth background) only overlaps P-BG (prediction background), (0 error), the GT component GT-1 overlaps P-BG (0 error), but P-BG overlaps both GT-BG and GT-1 (1 error). This behavior matches the behavior of the DIU and BM metric. Next, we consider the case where the size of two corresponding instances (surrounded by the background) in GT and prediction do not match perfectly, e.g., in some area, the prediction instance is larger than the GT instance, and in another area, the GT instance is larger. This would lead GT-BG to overlap P-BG and P-1 (1 error), GT-1 to overlap P-BG and P1 (1 error), P-BG to overlap GT-BG and GT-1 (1 error), and P-1 to overlap GT-BG and GT-1 (1 error). This is in stark contrast to the DIU and BM metrics. Both metrics yield an error of 0 as opposed to the calculated combined "false split" + "false merge" error of 4. Following A2, we must change the definition for counting "false splits" and "false merges" to $min(n, 1) - 1 - I$, where $I$ is a binary indicator that is 1 if and only if P-BG or GT-BG occurs in the list of corresponding labels. For the examples above, this results in the same behavior/error that we see in the BM and DIU metrics because an overlap with the background does not contribute to the loss. However, doing this would not allow the sum of "false merges" and "false splits" anymore to count the complete absence of an instance as described above. This is because "P-BG overlaps both GT-BG and GT-1" would suddenly result in an error of 0. This shows that depending on the choices that are made regarding handling the background, the "false splits" and "false merges" are similar to the DIU and BM error for some cases but differ in others.

Another interesting comparison is the case of multiple overlaps for the same instances. We consider Figure 10 (j)) as an example for this scenario. Here, the sum of "false merges" and "false splits" evaluates to 4 (follwing A1). Following A2, the error would be 0. In contrast to both these results, the DIU metric results in 2 because of an excess foreground component in the intersection and an excess background component in the union.

## A.10 QUALITATIVE RESULTS

Figure 14: Qualitative results for the cremi dataset.

## A.11 HYPERPARAMETER SEARCH SPACES

In Table 9, we show the search space of hyperparameters for each loss method for the Cremi dataset. The hyperparameters for each run are sampled randomly from the specified distributions. We do not use weight regularization for any model.

Table 9: Hyperparameter Search Space for Different Loss Methods on Cremi dataset

| Hyperparameter | Topograph | Dice | clDice | HuTopo | BettiMatching | Mosin |
|---|---|---|---|---|---|---|
| $\alpha$ | $[0.00001, 0.02]^{\beta}$ | N/A | $[0.001, 0.1]^{u}$ | $[0.0, 0.15]^{u}$ | $[0.0, 0.15]^{u}$ | $[0.001, 0.5]^{\beta}$ |
| ClDice alpha | N/A | N/A | $[0.1, 0.8]^{u}$ | N/A | N/A | N/A |
| $\alpha$ warmup epochs | $\{20, 50, 80\}$ | N/A | N/A | $\{20, 50, 80\}$ | $\{20, 50, 80\}$ | N/A |
| Learning rate | $[0.0001, 0.01]^{\beta}$ | | | | | |
| Channels | $\{[16, 32, 64, 128], [32, 64, 128, 256], [16, 32, 64, 128, 256], [32, 64, 128, 256, 512]\}$ | | | | | |
| Residual units | $\{2, 3, 4, 5\}$ | | | | | |
| Batch size | $\{8, 16, 32\}$ | | | | | |

$^{\beta}$: Log-uniform distribution, $^{u}$: Uniform distribution

## A.12 THRESHOLDING EFFECT

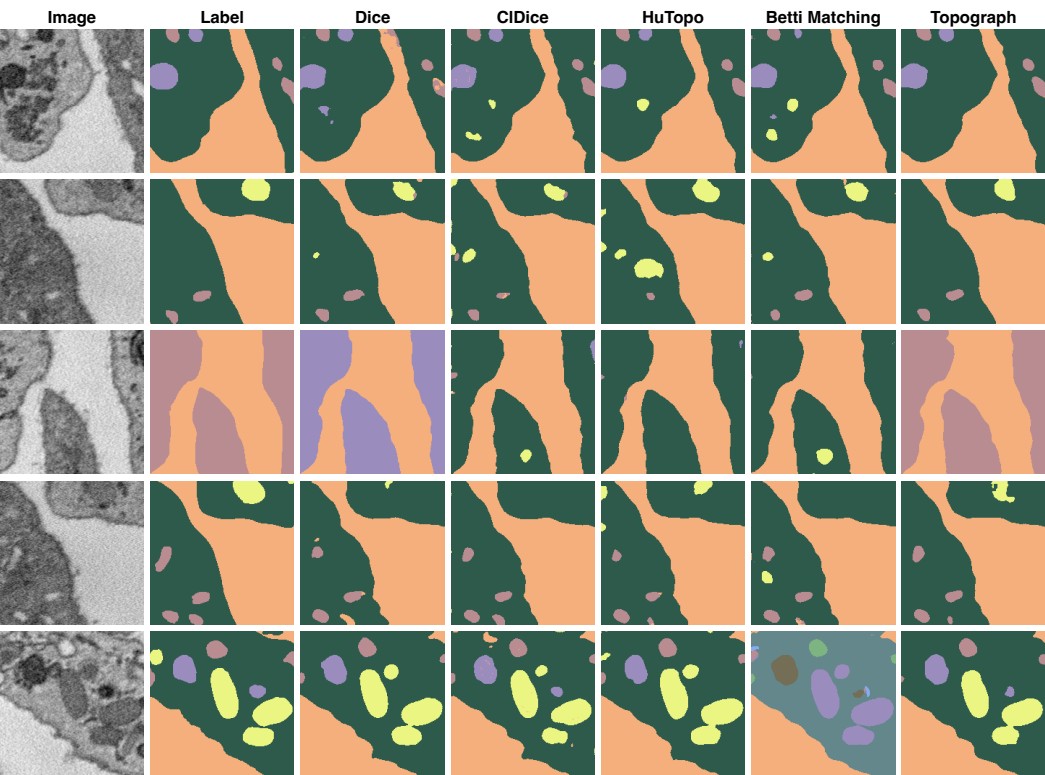

Figure 15: Qualitative results for the platelet dataset.

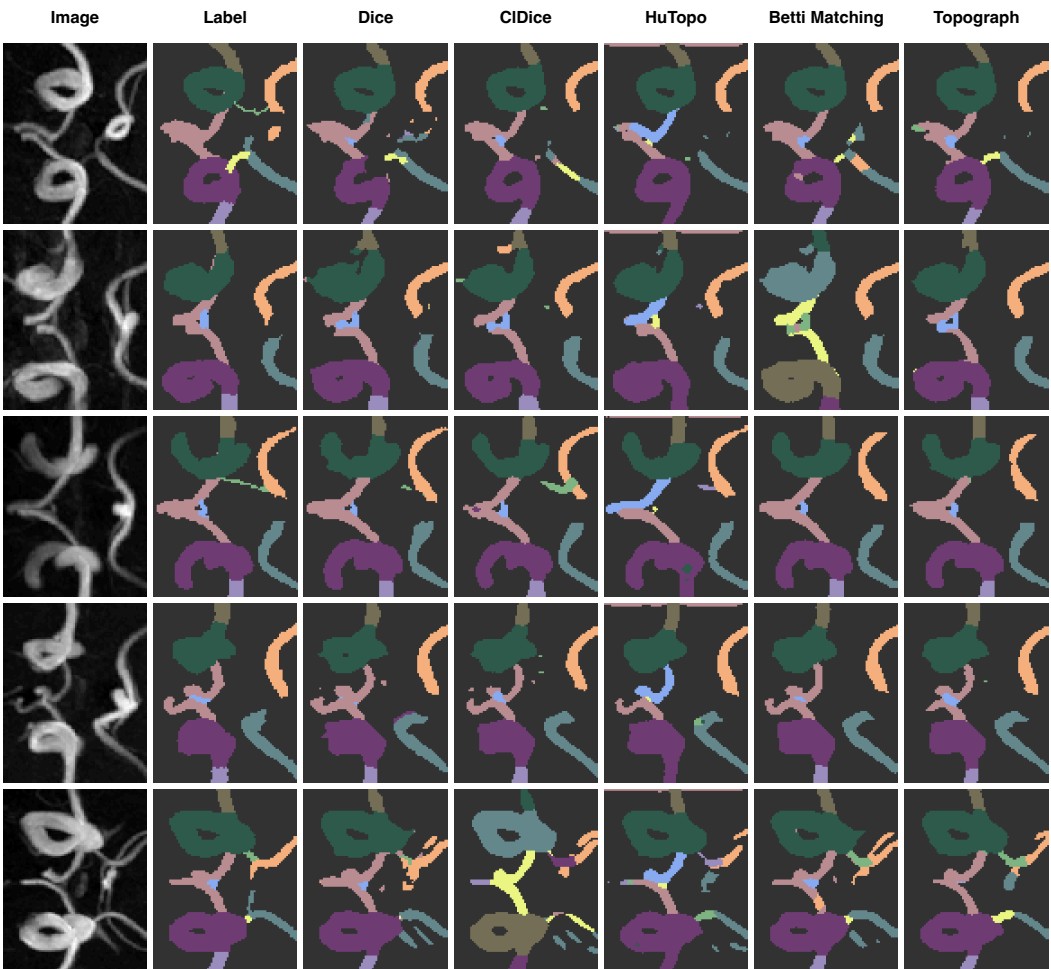

Figure 16: Qualitative results for the TopCoW dataset.

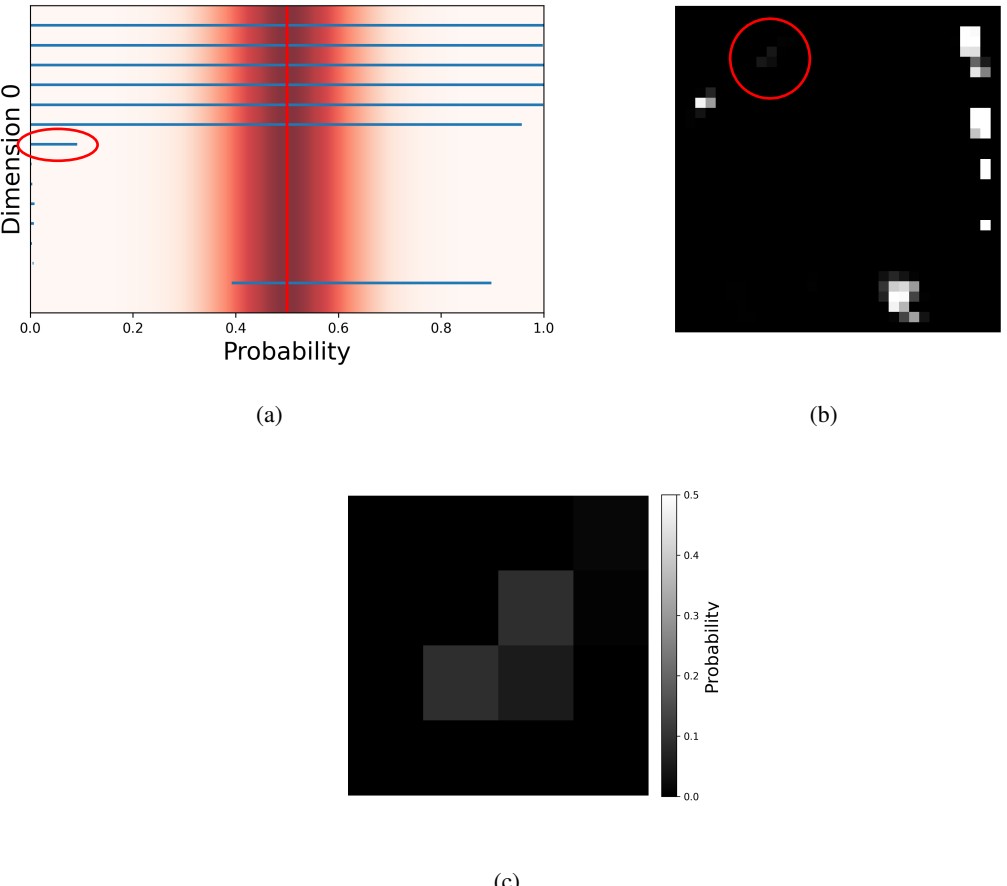

(a)

(b)

(c)

Figure 17: Intuitive illustration of the considered topological information. **(a)** Persistence barcode of dimension 1 for the image displayed in (b). **(b)** Output probability map of a segmentation network before binarization. **(c)** Crop of the component that is highlighted in (b) through a red circle. Here, we show a greyscale image and calculate the corresponding persistence diagram. All topological features that are "stabbed" by the red line (0.5 threshold) will be maintained and considered in our method. The endpoint of the interval in the red circle is distant from the binarization threshold and is, therefore, not considered a relevant topological structure for model training. The other disregarded features can be clearly interpreted as noise.

