# OpenReview forum: "Topograph: An Efficient Graph-Based Framework for Strictly Topology Preserving Image Segmentation"
_ICLR.cc/2025/Conference — ICLR 2025 Spotlight_

### Official Review · Reviewer_qvYt · 2024-11-03

**Soundness:** 3
**Presentation:** 3
**Contribution:** 3
**Rating:** 8
**Confidence:** 4

**Summary:**

The authors present a novel loss and a novel error metric for topology-aware image segmentation. The loss reflects homotopy equivalence between union and intersection of predicted and ground truth-segmentation. It is computed by establishing a component graph on (thickened/thinned) TP, FP, TN and FN components, and identifying "regular" vs non-regular nodes in the graph, where non-regular nodes point to topology-critical, spatially meaningful pixels in the prediction which are then penalised by the loss. Computation is 3-6 times faster than for related works on persistent homology-based losses.

**Strengths:**

The work provides an efficient loss that is nevertheless formally grounded, with strong topological guarantees. A comprehensive evaluation shows its strengths over existing approaches. Rich figures (also in the supplement) greatly facilitate the read.

**Weaknesses:**

-- Some definitions are not comprehensive and some Figures appear to be not in line with the text, making the work hard to grasp (see Questions)
--> fixed post-rebuttal

-- A more thorough discussion of some related work might render the work significantly more insightful (see Questions)
--> fixed post-rebuttal

-- The authors do discuss the 2d nature of the presented approach as a limitation. However, to clearly convey the significance of this limitation, it should be mentioned in this context that their evaluation features some 3d data (evaluated as 2d slices) -- here, imposing a 2d topological loss on slices appears unsuitable / overly strict.

**Questions:**

Re clarity:

-- "Nontrivial" intersection of closures (p. 4 l. 190) needs to be precisely defined (i.e., please provide a formal definition) -- I guess it has to mean "not finite", otherwise there could be edges between TP and TN and the component graph would not be bipartite? This confusion is furthered by Fig. 2 (see below) where such edge is actually present (so does "nontrivial" mean non-empty after all? but how is the comp. graph bipartite then?); Also, the respective edge is not present in Fig. 3. Please clarify.

-- In Fig. 2 (bottom right) the prediction foreground appears double-thickened, while later you settle for double-thickening of ground truth foreground; this causes an edge between intersection and background in Fig. 2 which should not be there and causes confusion; The same edge is also present in Fig. 7.3 -- I do not understand why; Same holds for the edge from the rightmost (thin) FN component to the spurious component in Fig. 7.3, I do not understand why this is there -- shouldn't it go from the FN component directly to the background node?

Re rel. work:

-- E.g., how is the DIU metric related to the common practice of counting "false split" and "false merge" errors, as commonly done for segmentation problems (cf [1] for one out of many examples where these are evaluated)? Could it be that DIU is the sum of false splits and false merges (at least for dim. 0 features?)? If not, do you see any other relation?

-- A discussion of [1] as related work appears warranted

-- Funke et al. (2018) is listed in rel. work as having a PH-based loss function, but this is not correct? Please discuss in more detail the relation of your work to their loss, which also focuses on topologically critical regions that are spatially correct

[1] https://doi.org/10.1016/j.ymeth.2016.12.013

Minor:

-- A pointer to Fig. 11 earlier in the text would be very helpful, latest on page 6 line 297 (in addition to pointing to Fig. 3)

---

> ### Author Response · Authors · 2024-11-20
>
> We thank you for your in-depth analysis of our proposed method and for the interesting references. In the following, we address your questions, focusing on (1.) Clarification of the definition of the combined component graph and (2.) Comparison to related work in the field of neuroscience and connectomics.
>
> ## Clarification of the definition of the combined component graph
>
> ### _The following text contains our initial answer, which had to be corrected in parts. We fixed this issue with the kind support of Reviewer qvYt; please read our later comment for the corrected answer._
>
> In fact, what we initially meant by "nontrivial" is that the intersection is "nonempty". We adjusted the formulation "nontrivial intersection" to "nonempty intersection," as suggested, which helps the comprehensibility of the paper. We understand the confusion caused by the edge representations in Fig. 2, Fig. 3, and Fig. 7. In fact, in Fig. 7, the edge from the rightmost FN component to the spurious component is a mistake that we missed. Here, this edge should not be present; instead, the edge should go directly to the background, as you pointed out. We corrected this mistake and updated the manuscript (see Fig 7.). As for Fig. 2, you are correct that an edge between the TP and TN does not exist. The cause of the confusion is that we used a representation with a different thickening. We replaced this with the correct thickening, as already displayed in Fig. 7. In Fig. 3, the representation of the component graph is correctly displayed without the presence of TP-to-TN edges. We are very grateful for your close examination of these details, which we have corrected now. We would appreciate hearing your feedback if this clarifies the definition of the combined component graph or if there are any ambiguities that we can address.

---

> > ### Comment · Reviewer_qvYt · 2024-11-25
> > **Re clarification of component graph:**
> >
> > I have one question that remains: don't you, in general, have a finite number of points in the intersection of (closure of) True Positives and (closure of) True Negatives, namely where a true positive region neighbours a false positive region? E.g., in the corrected Fig. 2, four points in the intersection of TPs and TNs? (That's why I was wondering if you had meant non-trivial = infinite)

---

> > > ### Author Response · Authors · 2024-11-25
> > >
> > > Once again, thank you for your valuable feedback and for revising your score. We want to apologize for the confusion and now understand the issue you raised: In certain cases, such as in Figure 2, we have a finite (but non-empty) intersection between the closure of TP and the closure of TN. Specifically, the intersection, in this case, includes the four points you identified. Our topological guarantees rely on G(P, G) being a bipartite graph whose edges only occur between vertices contained in T = TP ∪ TN and F = FP ∪ FN, as stated in Line 194. You are correct that this requires the introduction of edges if and only if the intersection between closures is non-finite (1-dimensional) as opposed to non-empty (also 0-dimensional). Unfortunately, we have missed this in our initial reply to your review. We have updated Line 190, Figures 2 and 7, and added a clarifying remark in our earlier answer. We sincerely acknowledge your detailed observations and constructive input, which have helped us improve the quality of our work.

---

> ### Author Response · Authors · 2024-11-20
>
> ## Comparison to related work
> _**R qvYt:** How is the DIU metric related to the common practice of counting "false split" and "false merge" errors?_
>
> _**TLDR Answer:** There are commonalities, but the metrics differ clearly in some cases. We have added a comprehensive discussion to the manuscript._
>
> A comparison with the practice of counting "false merges" and "false splits" turns out to be very interesting. To make a meaningful comparison with this practice, we have to make some prior assumptions. First, we assume that a "false split" is defined by a spatial overlap of two instances in the prediction with a single instance in the ground truth. A "false merge" is then defined by two instances of the ground truth being overlapped by a single instance in the prediction. Moreover, we assume that each instance resembles exactly one connected component with a unique label. To make a comparison to our method, we furthermore assume that there is a background class that shares a common label, but the background can be made up of one or more connected components (but is considered a single instance with a single label). Using the definitions in [1], the total number of false splits can be calculated by summing the value n-1 for every ground truth instance, where n is the cardinality of the set of spatially overlapped prediction labels. Similarly, the total number of wrong splits can be calculated by summing the value n-1 for every prediction instance, where n is the cardinality of the set of spatially overlapped ground truth labels.
>
> Furthermore, we consider two different options for handling the background. We are not certain which one is common practice in connectomics and would appreciate your response on that matter. First, the background can be viewed as a normal instance and thus contribute equally to the error calculation (A1). Second, the background can be ignored when calculating the error (A2). In the following paragraph, we investigate the effect of these background choices on different error cases.
>
> With the definitions made above and following A1, we now compare how the sum of "false splits" and "false merges" is related to the DIU metric. First, we consider the basic cases of having an instance in the GT and no corresponding instance in the predictions or reversed. Summing up the "false merges" and the "false splits" would result in an error of 1. We can calculate the result as follows: the GT-BG (ground-truth background) only overlaps P-BG (prediction background), (0 error), the GT component GT-1 overlaps P-BG (0 error), but P-BG overlaps both GT-BG and GT-1 (1 error). This behavior matches the behavior of the DIU and BM metric. Next, we consider the case where the size of two corresponding instances (surrounded by the background) in GT and prediction do not match perfectly, e.g., in some area, the prediction instance is larger than the GT instance, and in another area, the GT instance is larger. This would lead GT-BG to overlap P-BG and P-1 (1 error), GT-1 to overlap P-BG and P1 (1 error),  P-BG to overlap GT-BG and GT-1 (1 error), and P-1 to overlap GT-BG and GT-1 (1 error). This is in stark contrast to the DIU and BM metrics. Both metrics yield an error of 0 as opposed to the calculated combined "false split" + "false merge" error of 4.
>
> Following A2, we must change the definition of "false splits" and "false merges" to $min(n, 1) - 1 - I$, where $I$ is a binary indicator that is $1$ if and only if P-BG or GT-BG occurs in the list of corresponding labels. For the examples above, this results in the same behavior/error that we see in the BM and DIU metrics because an overlap with the background does not contribute to the loss. However, doing this would not allow the sum of "false merges" and "false splits" anymore to count the complete absence of an instance as described above. This is because "P-BG overlaps both GT-BG and GT-1" would suddenly result in an error of 0. This shows that depending on the choices that are made regarding handling the background, the "false splits" and "false merges" are similar to the DIU and BM error for some cases but differ in others.

---

> ### Author Response · Authors · 2024-11-20
>
> Another interesting comparison is the case of multiple overlaps for the same instances. We provide an example below, where **_0_** indicates a background class for GT and prediction, **_1_** is a ground truth foreground instance, and _**a**_ is a predicted foreground instance. If we use the initial definition (A1), the sum of "false merges" and "false splits" evaluates to 4. Following A2, the error would be 0. In contrast to both these results, the DIU metric results in 2 because of an excess foreground component in the intersection and an excess background component in the union. This is the error where multiple disconnected spatial overlaps of the same connected components occur, e.g., as displayed in Fig. 11.
>
> &nbsp; &nbsp; &nbsp; **GT:** &nbsp; &nbsp; &nbsp; &nbsp; &nbsp; &nbsp; &nbsp; &nbsp; **P:**
> **0 0 0 0 0** &nbsp; &nbsp; &nbsp; &nbsp; **0 0 0 0 0**
> **0 1 1 0 0** &nbsp; &nbsp; &nbsp; &nbsp; **0 0 a a 0**
> **0 1 0 0 0** &nbsp; &nbsp; &nbsp; &nbsp; **0 0 0 a 0**
> **0 1 1 0 0** &nbsp; &nbsp; &nbsp; &nbsp; **0 0 a a 0**
> **0 0 0 0 0** &nbsp; &nbsp; &nbsp; &nbsp; **0 0 0 0 0**
>
>
> &nbsp; &nbsp; &nbsp;&nbsp; &nbsp;&nbsp; &nbsp; **Comb:**
> &nbsp; &nbsp;&nbsp; &nbsp;**00 00 00 00 00**
> &nbsp; &nbsp;&nbsp; &nbsp;**00 10 1a 0a 00**
> &nbsp; &nbsp;&nbsp; &nbsp;**00 10 00 0a 00**
> &nbsp; &nbsp;&nbsp; &nbsp;**00 10 1a 0a 00**
> &nbsp; &nbsp;&nbsp; &nbsp;**00 00 00 00 00**
>
>
> Summarized, there are some very interesting commonalities between both metrics, but they behave differently in some important cases. Furthermore, the definition of "false merges" and "false splits" is highly dependent on the handling of the background class.
> In practice, the choice of the ideal metric for, e.g., a specific downstream application is highly task-dependent, and it is unlikely that a "one metric fits all" solution exists. However, we believe that the strict and general DIU metric, with its strong theoretical foundation, is an important addition to the repertoire of topological metrics in image segmentation.
>
> _**R qvYt:** A discussion of [1] as related work appears warranted. Funke et al. (2018) is listed in rel. work as having a PH-based loss function, but this is not correct?_
>
> _**TLDR Answer:** The missing related work is added to the main paper and to parts of the supplement. The citation of Funke et al. (2018) is corrected now._
>
> We thank you for pointing out the missing work from the field of connectomics and neuron segmentation and included this in our related work section. Regarding the TED metric discussed in [1], we included the discussion of the similarity to the "false splits" and "false merges" in the supplementary material. This should describe the special case of the TED metric where the boundary shift tolerance is 0. Explicitly, linking the relationship of the metrics with other boundary changes $>$ 0 is not meaningful because _any_ error that is completely contained within the tolerance distance of a boundary is not counted, while _any_ error that exceeds the tolerance distance is counted. In Figure 12, we added an interesting comparison to the TED metric with a boundary shift tolerance of 0 on some practical examples. We defined the metric following convention A1, where the background is an instance with a unique label that might be disconnected (e.g., multiple connected components). Regarding the foreground, we describe every single connected component as an independent instance and assign them unique labels. We believe that this is the most sensible comparison; however, we would greatly appreciate your feedback on the background convention commonly used within the connectomics community to ensure the comparison is as meaningful as possible.
>
>
> # Other
> _**R qvYt:** It should be mentioned in this context that their evaluation features some 3d data evaluated as 2d slices._
>
> We added information about using 2D projections/slices from 3D datasets in the paper. We invite you to read our discussion of a 3D generalization as a part of our answer to reviewer 1.
>
> _**R qvYt:** A pointer to Fig. 11 earlier in the text would be very helpful, latest on page 6, line 298 (in addition to pointing to Fig. 3)_
>
> We agree with your comment that this figure is crucial for understanding and now introduce the reference to the Figure in line 298.

---

> > ### Comment · Reviewer_qvYt · 2024-11-25
> > **Re comparison to related work**
> >
> > The discussion you have added to the paper, in particular regarding relation of your metrics to "false splits" and "false merges", is extremely interesting and imo boosts the impact of the work.
> > I'm increasing my score accordingly.

---

### Official Review · Reviewer_cmv9 · 2024-11-04

**Soundness:** 4
**Presentation:** 3
**Contribution:** 3
**Rating:** 8
**Confidence:** 3

**Summary:**

This paper presents a method that improves the topological consistency of segmentation while preserving overall performance, like IoU. Besides, a more topologically precise metric is also introduced in this paper.

**Strengths:**

1. The method proposed in this paper demonstrates a better solution for tasks emphasizing topological accuracy.
2. Besides, a new metric aims at topology consistency is also presented showing it's advantage over previous ones.
3. With a lower asymptotic complexity, the loss introduced in this paper can be computed in linear time which make the loss easy to implement.

**Weaknesses:**

1. The proposed method works well with relatively small datasets and UNet with fewer parameters. My concern is that if we have a larger dataset, can we achieve similar performance with a larger model without the implementation of Topograph?
2. In terms of binarization, this paper suggests that introducing a small random value would be helpful. But could it be beneficial if applied automatic thresholding like Otsu method?
3. In terms of DICE, Topograph seems to have similar performance with other methods. Is that indicating a tradeoff between topological critical pixels and topological irrelevant pixels?

**Questions:**

See Weakness.

---

> ### Author Response · Authors · 2024-11-20
>
> We thank you for your interest in our work and your questions regarding (1.) the utility of topological losses in data-rich scenarios, (2.) the effect of using automatic thresholding approaches, (3.) the presence of indicators for a tradeoff between topologically critical pixels and topologically irrelevant pixels.
>
> _**R cmv9:** The proposed method works well with relatively small datasets and UNet with fewer parameters. My concern is that if we have a larger dataset, can we achieve similar performance with a larger model without the implementation of Topograph?_
>
> Generally speaking, using more data is always beneficial for performance. We expect this behavior for pixel-wise metrics as well as for topological metrics. If there is enough high-quality data to achieve perfect pixel-wise accuracy (i.e., a Dice score of 1), there is no need for our Topograph method. Theoretically, a perfect Dice score also results in ground truth and prediction being identical, which implicitly fulfills the strictest topological guarantees. In practice, this does not occur frequently, and arguably most methodical improvements are useless under this scenario. Thus, we try to answer the questions if our method is still useful if we _increase_ the amount of data used for the training. We performed an ablation experiment to get an intuition on how to answer this question. The table below shows the results of an ablation of the dataset size. We found that our method improves performance across all the different dataset sizes. Furthermore, the ablation results do not indicate that the relative topological performance improvement gets smaller with an increased amount of data. We are interested in investigating this question further and planning further experiments with other datasets and additional hyperparameter tuning of the models.
>
> | **Dataset. Frac.** | **BM ↓**  ($\mathcal{L}\_{CG}$)    | **DIU ↓**  ($\mathcal{L}\_{CG}$)     | **BM ↓**    ($\mathcal{L}\_{Dice}$)  | **BM ↓**    ($\mathcal{L}\_{Dice}$)   | **BM Improvement** | **DIU Improvement** |
> |------------------|----------------|----------------|----------------|----------------|----------------|----------------|
> | 12.5%              | 72.8929       | 86.5714        |76.3333       | 89.1667        | 4.51%    | 2.91%
> | 25.0%              | 72.5357       | 83.9286        |74.8095       | 88.3961        | 3.04%    | 5.05%
> | 37.5%              | 68.5238       | 84.5952        |75.5238       | 89.6071        | 9.27%    | 5.59%
> | 50.0%              | 66.7857       | 84.0952        |67.4881       | 86.1191        | 1.04%    | 2.35%
> | 62.5%              | 65.2143       | 82.8333        |65.8095       | 83.3333        | 0.90%    | 0.60%
> | 75.0%              | 62.9048       | 79.9048        |64.6191       | 80.0238        | 2.65%    | 0.15%
> | 87.5%              | 62.2738       | 81.2976        |65.4881       | 82.0476        | 4.91%    | 0.91%
> | 100%               | 59.2381       | 76.4167        |64.3333       | 80.9524        | 7.92%    | 5.60%

---

> ### Author Response · Authors · 2024-11-20
>
> _**R cmv9:** In terms of binarization, this paper suggests that introducing a small random value would be helpful. But could it be beneficial if applied automatic thresholding like Otsu method?_
>
> _**TLDR Answer:** We experimented with the Otsu method and found that it is effective but not always better than random variation._
>
> Thank you for suggesting the Otsu method as an alternative approach for threshold selection, it is an interesting approach that we have not yet experimented with. Please also refer to the answer to Reviewer U2U3, where we discussed our intuition and experimental results on our thresholding choices in detail. We implemented the Otsu method into our method and included the results in Table 3 of the paper (the table is also displayed below), which contains the results for our ablation on the threshold variation parameter. The ablation showed that using the Otsu threshold is favorable compared to using a fixed threshold at 0.5 or using a suboptimal threshold variation. However, we still achieve the best results using threshold variation when choosing an optimal parameter. In our new experiments, we see that the Otsu threshold of the probability maps lies close to the binarization threshold of 0.5. This is expected as the minimal dice loss is achieved when the background and foreground are at 0 and 1.
>
>
> | **Thres. Var.** | **DIU ↓**      | **BM ↓**       | **Dice ↑**    |
> |------------------|----------------|----------------|---------------|
> | 0.0              | 51.5625       | 40.6250        | 0.80886       |
> | 0.01             | 47.2500       | 37.7500        | 0.81965       |
> | 0.05             | **46.1250**   | _36.2500_      | 0.79447       |
> | 0.1              | _46.7500_     | **35.5625**    | **0.82555**  |
> | 0.2              | 47.8750       | 38.2500        | 0.80861       |
> | 0.5              | 49.2500       | 36.5625        | _0.82431_     |
> | Otsu             | 49.1250       | 37.0000        | 0.82396       |
>
>
> In conclusion, we see automatic thresholding as an interesting option or hyperparameter in our method, e.g. in resource-limited settings where one wants to reduce the hyperparameter search space as far as possible. We appreciate your suggestion and will incorporate automatic thresholding as a hyperparameter in our GitHub repository.
>
> _**R cmv9:** In terms of DICE, Topograph seems to have similar performance with other methods. Is that indicating a tradeoff between topological critical pixels and topological irrelevant pixels?_
>
> _**TLDR Answer:** Our experiment results do not indicate that such a tradeoff exists._
>
> Based on our experimental results, we do not see conclusive evidence for a tradeoff between correcting topologically critical and topologically irrelevant pixels. Our results indicate that, by using our method, it is possible to significantly improve topological accuracy. Given that observation, we must have an increased number of correct _topologically critical pixels_. Second, our results show that our method does not significantly improve or deteriorate the Dice score. Here, it is important to stress that a significant improvement of topological accuracies can be achieved by a marginal number of changed pixels compared to the total number of pixels in an, e.g., a 375x375 image with 140625 pixels. It can be enough to fix a handful of pixels (this would be less than 0.02% of the total number of pixels) to achieve significant improvements in terms of topological correctness. Consequently, even if no tradeoff exists — meaning all topologically relevant pixels contribute positively to the Dice score without introducing additional topologically irrelevant errors — we would not expect to observe significant Dice score improvements because the number of additional correct pixels is very small. These additional correct pixels are not distinguishable from noise, e.g., caused by hyperparameter tuning. Therefore, we do not see any conclusive evidence that such a tradeoff exists. Similarly, we do not have conclusive evidence for the absence of such a tradeoff. However, our results show that any hypothetically existing tradeoff has no significant effect on the Dice performance, which is an advantage of our method.

---

### Official Review · Reviewer_8zuR · 2024-11-09

**Soundness:** 2
**Presentation:** 2
**Contribution:** 2
**Rating:** 6
**Confidence:** 2

**Summary:**

The authors present a method to address the segmentation problem with a focus on preserving the topology of segmented regions. They also introduce a metric for evaluating the accuracy of the predicted segmentation. The proposed approach consists of several stages: (1) The input image is binarized and overlapped with ground truth segments; (2) A graph is constructed based on this overlapped image; (3) Superpixels are created, with each node in the graph representing a superpixel; (4) A set of misclassified nodes is identified, and nodes that do not impact topological structure are removed; (5) Optimization is performed only on the nodes in this remaining set. The authors evaluate their method on multiple datasets using a range of metrics.

**Strengths:**

The proposed method seems to preserve the topology of segmented regions, addressing a challenge in segmentation tasks.

**Weaknesses:**

The minimization problem formulation is not well-defined (objective function, parameters, and regularization).

**Questions:**

How does the method handle edge cases with the $\alpha$ parameter? Specifically, what is the outcome if $\alpha$ is set to 0 or to a very large value?

---

> ### Author Response · Authors · 2024-11-20
>
> We thank you for the feedback and address your questions below. We hope that our explanations below can improve the presentation and clarity of our method.
>
> _**R 8zuR:** The minimization problem formulation is not well-defined (objective function, parameters, and regularization)._
>
> Our final minimization problem consists of our topological loss $\mathcal{L}_{CG}$ (Equation 4) combined with a pixel-wise loss, such as the Dice loss (Section 2.3), as shown in the equation below. However, the pixel-wise loss function can be chosen arbitrarily and is not limited to the Dice loss function. We provide the complete hyperparameter search space in Appendix A.11. We do not use weight regularization and added this information to our hyperparameters. We hope this answers your question and are happy to answer any further questions you may have.
>
> $$
> \mathcal{L}=\mathcal{L}\_{Dice}+\mathcal{L}\_{CG} = \mathcal{L}\_{Dice} + \alpha \sum\_{v \in \mathcal{V}\_c} \bar{y}\_v
> $$
>
> _**R 8zuR:** How does the method handle edge cases with the $\alpha$ parameter? Specifically, what is the outcome if is set to 0 or to a very large value?_
>
> Concerning the edge cases of the $\alpha$ parameter, we have conducted a comprehensive ablation study (detailed in Appendix A.5) that specifically addresses these scenarios. When $\alpha = 0$, the model relies solely on the pixel-wise loss, resulting in topologically inferior segmentations compared to our combined approach. All Dice loss results in Table 1 resemble the edge case of alpha being zero. With higher values for $\alpha$, the Topograph loss completely dominates the pixel-wise loss, leading to unstable training behavior. This is evidenced in Table 4, where experiments with high $\alpha$ values achieved their best performance during the warm-up phase before the topological loss was activated. As an additional insight, we included a visualization of an unstable training run in Figure 13.

---

> > ### Comment · Reviewer_8zuR · 2024-11-26
> >
> > Thank you for your response. I will keep my score.

---

### Official Review · Reviewer_U2U3 · 2024-11-09

**Soundness:** 4
**Presentation:** 3
**Contribution:** 3
**Rating:** 8
**Confidence:** 4

**Summary:**

This paper proposes a new loss function, called **Topograph**, for image segmentation, which aims to preserve the topological accuracy of predictions. The authors highlight the importance of **topological correction** in many segmentation tasks, especially in the medical domain where it is crucial for accurate diagnosis and functional analysis. The approach relies on the construction of a component graph that encodes the topological information of both the ground truth and the prediction.

The main innovations are:
* **A DIU (Discrepancy between Intersection and Union) metric:** This new metric captures topological correctness with strict theoretical guarantees, notably by capturing the homotopy equivalence between the union and the intersection of a label/prediction pair. The DIU metric is more sensitive to fine-grained topological differences than existing metrics such as the Betti number error and the Betti matching error.
* **A graph-based loss function:** This general loss function preserves topology and can be used to train various segmentation networks. It is based on a component graph that combines topological information from the ground truth image and the prediction.
* **Computational Efficiency:** Topograph outperforms existing methods in terms of topological correction of predictions while being efficient in terms of time and resources due to its low asymptotic complexity (O(n α(n))).

**Strengths:**

0. The paper is clearly written and easy to understand.

1. The paper presents **theoretical guarantees** for Topograph, demonstrating that if their novel introduced loss is zero, then there is homotopy equivalence. This means that Topograph not only guarantees homotopy equivalence between the ground truth and the segmentation, but also between their union and intersection, thus capturing the spatial correspondence of their topological properties.

2. The approach has a lower complexity than persistent-homotopy (PH) based ones, and, experimentally, a faster running time.

3. Experimental results show that Topograph improves topological accuracy compared to pixel-based loss functions and other topology-preserving approaches, as shown by the best scores on DIU and BM metrics. At the same time, it maintains pixel-level accuracy comparable to the best benchmarks, indicating that there is no trade-off between pixel-level accuracy and topological correctness.

**Weaknesses:**

1. The binarization used in Topograph may result in a loss of topological information compared to PH-based methods. It might be possible to extend the construction of the component graph similarly to what is done with component trees (as defined in mathematical morphology), but it is not straightforward to do so.

2. The method is currently limited to 2D images. Its extension to 3D images is considered, but I believe it is not obvious to do. The component graph as currently defined is difficult to extend to 3D.

**Questions:**

Although the method demonstrates superior results in 2D, I am not convinced that it will be easy to extend to 3D, and I would like to understand the authors' intuition and thoughts about a 3D extension. The authors may want to discuss some specific challenges they anticipate in extending Topograph to 3D, and to outline potential strategies they are considering to address these challenges.

Regarding the binarization, I am wondering if it is possible to quantify or estimate how much topological information is lost through
the binarization approach compared to PH methods. Is it possible to suggest some strategy that mitigate this information loss while maintaining Topograph computational efficiency advantages?

---

> ### Author Response · Authors · 2024-11-20
>
> We thank you for your interesting questions. In the following, we address the questions that focus on the raised topics: (1.) Comparison to PH-based methods and (2.) Generalization to the 3D case.
>
> ## Comparison to PH-based methods
> _**R U2U3:** Is it possible to quantify or estimate how much topological information is lost through the binarization approach compared to PH methods?_
>
> _**TLDR Answer:** Yes, we can exactly quantify the lost information using concepts from PH, but our experiments indicate that this information is not crucial for training a segmentation model._
>
> As you pointed out, the binarization leads to a loss of topological information. Using persistent homology, we can exactly quantify the amount of information that is lost by considering an image's persistence barcode representing topological features in dimension 0 and dimension 1. In such a barcode, the threshold can be interpreted as a vertical stabbing line for the persistence intervals in a barcode (see Figure 17 in the Supplement). The lost information contains all persistence intervals that are not stabbed by this vertical line, i.e., all topological features having a persistence interval that does not include the binarization threshold (e.g. 0.5).
>
> _How important is the lost topological information for training a topology-preserving segmentation neural network?_
>
> _**TLDR Answer:** Our experiments indicate that our approach does not lose important information because the information around the binarization threshold is most important for model training._
>
> While the complete topological information (as captured by PH) is interesting for topological data analysis, our experiments indicate that, for topology-preserving image segmentation, it is mostly the topological information _around_ the typical binarization threshold of 0.5 that is crucial for network training. Experimentally, we explored this property by introducing the threshold variation parameter. We found that varying the binarization thresholds during training (the final 0.5 threshold in the evaluation is constant) enhances the performance of our models compared to training only with a fixed threshold (see Table 3). However, allowing the threshold across the full range of [0, 1] did not improve performance compared to focusing on a more targeted region around 0.5, e.g., by setting the threshold variation parameter to 0.05, which results in $\sim 95$% of the random binarization threshold being in the range of [0.4, 0.6]. We interpret these results as follows: thresholding only at 0.5 is bad for performance because topological features that occur close to the threshold but do not pass through it are ignored. During inference, these features can easily "slip" into the threshold area and cause topological errors in the predicted binarized segmentation. Secondly, focusing too much on features that only exist far from the 0.5 binarization threshold does not help the model training. Our intuition is that focusing on these outlying persistence intervals does not benefit the final performance since these features are too far away to slip into the binarization threshold area in the inference stage. Furthermore, most of these features correspond to noise in the likelihood map, and therefore, they are not as relevant for topological optimization; see more detailed discussion below and Figure 17.
>
> We added an example in the supplement (Figure 17) for an intuitive illustration of the considered topological information. There, we show a greyscale image and calculate the corresponding persistence diagram. All topological features that are "stabbed" will be maintained and considered in our method.
>
> The endpoint of the interval in the red circle is distant from the binarization threshold and is, therefore, not considered a relevant topological structure for model training. The other disregarded features can be clearly interpreted as noise.
>
>
>
> _**R U2U3:** Is it possible to suggest some strategy that mitigates this information loss (compared to PH) while maintaining Topograph computational efficiency advantages?_
>
> _**TLDR Answer:** Yes, we mitigate the effect of this information loss on the training by varying the threshold for the same sample across epochs._
>
> Regarding your question concerning computation efficiency, we believe that the threshold variation is achieving exactly this: It acquires information from multiple points in the persistence diagrams by thresholding at different values in different epochs but stays efficient because, at every single training iteration, only the binarized signals are evaluated. We are very interested in generalizing our approach to the persistence homology level, e.g., to allow for more interesting insights using topological data analysis. We are thankful for your suggestion regarding the component trees and will try to experiment with them.

---

> ### Author Response · Authors · 2024-11-20
>
> ## Extension to the 3D case.
> _**R U2U3:** I would like to understand the authors' intuition and thoughts about a 3D extension._
>
> _**TLDR Answer:** Extending the component graph to 3D is possible, and we believe that it can be a useful method for datasets that are governed by their topological complexity in dimension 0 and dimension 2. However, we anticipate that a simple generalization of the current approach fails to capture topological information in dimension 1._
>
> We can generalize the construction of the component graph as well as the combined component graph to 3D. For this extension, we will define connected components and neighborhoods for the 6-neighborhood and 26-neighborhood conventions for voxels. Based on this, we can also define the combined component graph proposed in our current work.
>
> _**R U2U3:** The authors may want to discuss some specific challenges they anticipate in extending Topograph to 3D._
>
> Although we can build the combined graph similarly in 3D, we anticipate that we can not provide the same strict topological guarantees because of issues in encoding the topological information in dimension 1. Specifically, the component graphs for the sphere and the torus will turn out exactly the same (a chain of BG-FG-BG), i.e., we cannot distinguish them.
>
> However, we believe that the extension to the 3D case is not without merits because it can still encode valuable information about connected components. Moreover, we believe that we can maintain some guarantees for the inclusions with intersection and union regarding connected components, i.e. topological features in dimension 0. By Alexander duality, information from features in dimension 2 can also be encoded. In practice, for the large number of datasets where connected components constitute the topologically most interesting parts of the domain (for example, cell microscopy), the 3D extension will still be highly relevant and open for future work.

---

> > ### Comment · Reviewer_U2U3 · 2024-11-25
> >
> > Thank you for the detailed answers.
> >
> > I maintain my score.

---

### Author Response · Authors · 2024-11-25

Dear Reviewers,

We would like to thank you all again for your thorough and helpful reviews; many interesting points were raised, and we genuinely believe our work has improved as a result of your suggestions. As the discussion period is ending soon, we would greatly appreciate the chance to discuss our changes with you and hope you find the time to respond to our rebuttal. If you believe we have already addressed your questions and concerns, we would, of course, appreciate an appropriate adjustment of your scores to reflect this.

Many thanks and best wishes,
the authors

---

### Meta-Review · Area_Chair_nxai · 2024-12-21

**Metareview:**

This paper proposes a topology-preserving loss for binary and multi-label image segmentation. Another contribution of this paper is a new metric for topology correctness. All four reviewers gave positive ratings of acceptance for this paper. The loss is technically sound and theoretically justified with a strong guarantee for topology preserving. In practice, using this loss does not introduce much computational overhead and can be beneficial for many segmentation networks and datasets as shown in the experiments. The loss is particularly useful in low-data regimes such as for medical images.
However, qualitative results don't show a significant difference of segmentation between the proposed method and other baselines such as using Dice loss. The AC encourages the authors to identify examples with more obvious improvement in terms of topology in the final version, and also include a discussion of extension to 3D segmentation as mentioned by the reviewer.

**Additional Comments On Reviewer Discussion:**

Reviewer U2U3 is concerned about the limitation to 2D. The authors have discussed possible extensions to 3D which is not straightforward. The AC believes that 3D extension can be left as future work and is not a must-have for this paper.

---

### Decision · Program_Chairs · 2025-01-22

Accept (Spotlight)